# In Situ Raman Analysis of Biofilm Exopolysaccharides Formed in *Streptococcus mutans* and *Streptococcus sanguinis* Commensal Cultures

**DOI:** 10.3390/ijms24076694

**Published:** 2023-04-03

**Authors:** Giuseppe Pezzotti, Satomi Ofuji, Hayata Imamura, Tetsuya Adachi, Toshiro Yamamoto, Narisato Kanamura, Eriko Ohgitani, Elia Marin, Wenliang Zhu, Osam Mazda, Azusa Togo, Satoshi Kimura, Tadahisa Iwata, Hideki Shiba, Kazuhisa Ouhara, Takashi Aoki, Toshihisa Kawai

**Affiliations:** 1Ceramic Physics Laboratory, Kyoto Institute of Technology, Sakyo-ku, Matsugasaki, Kyoto 606-8585, Japan; 2Department of Immunology, Graduate School of Medical Science, Kyoto Prefectural University of Medicine, Kamigyo-ku, 465 Kajii-cho, Kyoto 602-8566, Japan; 3Department of Orthopedic Surgery, Tokyo Medical University, 6-7-1 Nishi-Shinjuku, Shinjuku-ku, Tokyo 160-0023, Japan; 4Department of Dental Medicine, Graduate School of Medical Science, Kyoto Prefectural University of Medicine, Kamigyo-ku, Kyoto 602-8566, Japan; 5Department of Applied Science and Technology, Politecnico di Torino, Corso Duca degli Abruzzi 24, 10129 Torino, Italy; 6Department of Molecular Science and Nanosystems, Ca’ Foscari University of Venice, Via Torino 155, 30172 Venice, Italy; 7Department of Biomaterial Sciences, Graduate School of Agricultural and Life Sciences, The University of Tokyo, 1-1-1 Yayoi, Bunkyo-ku, Tokyo 113-8657, Japan; 8Department of Biological Endodontics, Graduate School of Biomedical and Health Sciences, Hiroshima University, 1-2-3 Kasumi, Minami-ku, Hiroshima 734-8553, Japan; 9Department of Periodontal Medicine, Graduate School of Biomedical and Health Sciences, Hiroshima University, 1-2-3 Kasumi, Minami-ku, Hiroshima 734-8553, Japan; 10Faculty of Fiber Science and Engineering, Kyoto Institute of Technology, Sakyo-ku, Matsugasaki, Kyoto 606-8585, Japan; 11Department of Oral Science and Translational Research, College of Dental Medicine, Nova Southeastern University, 3301 College Ave, Fort Lauderdale, FL 33314, USA

**Keywords:** *Streptococcus sanguinis*, *Streptococcus mutans*, cocultures, in situ Raman spectroscopy, biofilm exopolysaccharides, antagonistic interactions

## Abstract

This study probed in vitro the mechanisms of competition/coexistence between *Streptococcus sanguinis* (known for being correlated with health in the oral cavity) and *Streptococcus mutans* (responsible for aciduric oral environment and formation of caries) by means of quantitative Raman spectroscopy and imaging. In situ Raman assessments of live bacterial culture/coculture focusing on biofilm exopolysaccharides supported the hypothesis that both species engaged in antagonistic interactions. Experiments of simultaneous colonization always resulted in coexistence, but they also revealed fundamental alterations of the biofilm with respect to their water-insoluble glucan structure. Raman spectra (collected at fixed time but different bacterial ratios) showed clear changes in chemical bonds in glucans, which pointed to an action by *Streptococcus sanguinis* to discontinue the impermeability of the biofilm constructed by *Streptococcus mutans*. The concurrent effects of glycosidic bond cleavage in water-insoluble α − 1,3–glucan and oxidation at various sites in glucans’ molecular chains supported the hypothesis that secretion of oxygen radicals was the main “chemical weapon” used by *Streptococcus sanguinis* in coculture.

## 1. Introduction

The concurrent effects of repeated subsistence/starvation cycles and sudden environmental pH fluctuations make the human oral cavity an environment of fierce bacterial competition [1]. In such a competitive environment, dental biofilm features a number of key interspecies interactions, which might suddenly alter oral microbiome homeostasis and cause diseases. Dictated by environmental conditions, typified by a high biodiversity and cellular density [2,3,4,5], and affected by a defenseless accessibility from external sources [6], biofilm is the chaotic arena of multiple and concurrent interspecies interactions leading to bacterial competition and coexistence. Oral biofilms are pathogenic in nature, and their excessive formation, in combination with a deregulated immune response, leads to intraoral diseases, such as dental caries, gingivitis, and periodontitis.

An important case of interspecies interaction taking place on the enamel surface is the antagonism between streptococcal bacteria [7]. Among supragingival streptococcal species, *Streptococcus sanguinis* (*S. sanguinis*) pioneers the colonization of symbiotic bacteria and is associated with oral health while *Streptococcus mutans* (*S. mutans*) is a major pathogen causing human dental caries and promoting the development of an acidogenic and aciduric environment that enhances pathogenicity from other organisms in the oral cavity. The concentration ratio of these two species has been regarded as an indicator of oral health [8,9]. The antagonism between *S. mutans* and *S. sanguinis* at the ecological level is long known [8,10], and a number of studies have phenomenologically featured the interaction between the two streptococcal bacteria as a function of time, inoculation sequence, and reciprocal concentrations [11,12,13]. The antagonistic interaction referred to as “competitive exclusion” between *S. mutans* and *S. sanguinis* was early demonstrated by altering the sequence of inoculation in germ-free rats [10]. Kreth et al. [11] showed that, upon inoculating bacteria in sequence, the early species inoculated inhibits growth of the later inoculated one while simultaneous inoculation of both species always led to coexistence. An important discovery of the above study was that each bacterial species attempted to repress the antagonist by using a “chemical weapon”: *S. sanguinis* produced hydrogen peroxide (H_2_O_2_) while *S. mutans* synthesized mutacin (a polycyclic peptide containing thioether amino acids). These findings were later supported by both clinical research and in silico analyses [14,15]. However, the machineries of mutual inhibition are not completely understood yet, and the environmental conditions affecting the ability of *S. sanguinis* to maintain an ecologically balanced biofilm in the oral cavity is awaiting further clarifications. Therefore, establishing a suitable method of molecular-scale visualization is essential in order to clarify interspecies interaction for this important bacterial duality. Visualizing and rationalizing the mechanisms behind interspecies interaction at the molecular level could open new paths toward the development of novel methods of diagnosing/curing oral biofilm formation.

Several authors have used vibrational spectroscopic techniques to characterize bacterial species and biofilm structures at the molecular level [16,17,18,19]. Infrared and Raman spectroscopies can both be used to acquire molecular-scale information on bacterial biofilms by exploiting bond vibrational fingerprints. The former technique is indeed widely used for this purpose, but interference with signals from water bonds (–OH) disturbs the scatter from key biofilm molecules, thus limiting its use to dehydrated biofilm samples [20]. On the other hand, Raman spectra (excited with visible light) are not affected by water signals and equally provide clear fingerprints of biological molecules with micrometric spatial resolution [21,22,23]. Since no dehydration or stain/markers are needed to obtain Raman spectra, Raman measurements can be performed in situ with a procedure compatible with bacterial life. In previous studies [23,24,25,26,27,28,29], we have performed Raman spectroscopic analyses of cells, bacteria, pathogenic yeasts, and viruses upon exploiting a machine-learning algorithm linked to a library of Raman spectra from elementary molecules. The analytical approach followed in this study is in line with those previous studies and represents a first attempt to their further extension to the analysis of bacterial biofilms.

In this study, we examined in vitro formed dental biofilm in *S. mutans* and *S. sanguinis* in separate cultures and in coculture, upon their simultaneous in vitro inoculations with intermediate fractional populations. Raman spectra were systematically collected on cocultures at a fixed time and processed through a deconvolution-based machine-learning algorithm to analyze and compare live biofilm metabolomics. Important new details could be unveiled regarding interspecies interactions picturing their coexistence under severe competition. The insight gained through in vitro Raman experimentation could become a guide in future clinical studies of patient samples.

## 2. Results

### 2.1. Raman Calibrations of Water-Insoluble Exopolysaccharides

An important preliminary step in the present investigation resided in quantitatively calibrating the dependence of the Raman spectrum of α–glucan on the structure of its O-glycosidic bonds. Figure 1a shows the structures and fingerprint Raman vibrational modes of α − 1,6– and α − 1,3–glucans, while (b) gives the respective spectra collected on pure compounds (cf. labels in inset; cf. band assignments as summarized in Appendix A).

The spectra differ regarding several important details, which relate to specific features in the respective bond structures. In particular, three spectral areas can be envisaged in Figure 1b, namely at 500~600 cm^−1^ (Zone I, henceforth), 900~950 cm^−1^ (Zone II), and 1450~1570 cm^−1^ (Zone III), which correspond to vibrational modes of C–C–O(H) deformation, C–O–C stretching, and partly to H–C–OH bending, respectively (cf. Figure 1a). Raman spectra collected on mixtures of elementary compounds with different fractions (α − 1,3– to α − 1,6–glucans ratios equal 0:1, 1:3, 1:2, 3:1, and 1:0) were compared in Zone I~III (Figure 2a–c, respectively). In Zone I (Figure 2a), the band located at ~555 cm^−1^ (C2–C1–O) is common to both α − 1,3– to α − 1,6–glucan structures, while the band located at ~520 cm^−1^ (C2–C3–OH) should only appear in the α − 1,6–glucan structure (cf. Figure 1a). Zone II (Figure 2b) includes bands at 919 and 948 cm^−1^, which are fingerprints of glycosidic C–O–C stretching peculiar to α − 1,6– and α − 1,3–glucan structures (i.e., C1–O–C6 and C1–O–C3, respectively) (cf. Figure 1a). In Zone III (Figure 2c), the band at ~1484 cm^−1^ is common to both α − 1,3– and α − 1,6–glucan structures (H–C6–H bending; cf. Figure 1a), while the band at 1568 cm^−1^ is only seen in the spectrum of α − 1,3–glucan. However, the presence of a band at ~1568 cm^−1^ is extrinsic to the glucan structure and cannot be explained without considering the presence of ring C=C bonds [30]. The presence of such bonds could be related to residues of glucansucrase enzyme used by *S. mutans* to synthesize the α − 1,3–glucan structure. This point will be discussed in more detail in a later section.

Variations of the Raman intensity ratios, *R*^(1)^_α3/α6_ = *I*_520_/*I*_555_, *R*^(2)^_α3/α6_ = *I*_948_/*I*_919_, and *R*^(3)^_α3/α6_ = *I*_1568_/*I*_1482_, are plotted as a function of sample compositions in Figure 3a–c, respectively. The obtained dependencies represent calibration plots that, in principle, enable us to independently quantify the α − 1,3– to α − 1,6–glucans ratio from Raman spectra collected in samples of unknown composition. The survivorship of these three different Raman methods for assessing the glucan structure of live biofilm will be tested in later sections. The fraction of α − 1,3–glucans, F_α3_, represents an important structural parameter because it links to the impermeability of the biofilm, as discussed later. An application of the Raman parameter, *R*^(2)^_α3/α6_ = *I*_948_/*I*_919_, to the quantification of the α − 1,3–glucan fraction in *Candida auris* biofilm has been described elsewhere [31]. 

### 2.2. Morphological and Structural Characterizations of Biofilms

Figure 4a–e show optical micrographs of concurrently inoculated cocultures (at 24 h) with *S. mutans*– to –*S. sanguinis* ratios of 1:0, 1:1, 1:4, 1:8, and 0:1, respectively. The micrographs show abundant biofilm formation whatever the initial bacterial fraction except for the culture of *S. sanguinis* alone, which showed a negligible amount of biofilm. OD values collected on different cocultures are shown in Figure 4f.

The OD data, which are collected on solutions in which both soluble and insoluble glucans were completely dissolved, represent a measure of the amount of biofilm formed. The data reveal an interesting trend by showing that the amount of biofilm developed by *S. sanguinis* alone was the lowest detected in this study (i.e., in agreement with microscopic observation) while the one formed in the 1:8 coculture (i.e., the richest in *S. sanguinis* fraction) was the highest as a result of an increasing trend with increasing fraction of *S. sanguinis* population. Note that the *S. sanguinis* ability to form biofilm greatly depends on growth medium [32,33], and the scarce ability of *S. sanguinis* to form biofilm in BHI medium, which is the case here, has already been reported by other authors [33]. It has also been suggested that *S. sanguinis* adheres to the extracellular glucan produced by other streptococci via putative glucan-binding proteins [34,35]. Although we cannot completely rule out the hypothesis that *S. mutans* (in its smallest fraction in coculture) could produce the most abundant amount of biofilm, the present data might rather suggest that *S. sanguinis* exploits environment/metabolites related to the presence of other streptococci to produce copious amounts of a biofilm suitable for its survivorship. In this context, note that the OD value only reveals the amount of biofilm formed while it gives no information about its chemical structure. This important point will be clarified in the next section.

Nuclear magnetic resonance (NMR) was applied to determine the chemical structure of the glucan in the biofilm of the *S. mutans* culture and to calculate the ratio of each glucan component (Figure 5). The NMR spectra of glucans in the biofilm are shown in Figure 5a while the reference spectra of α − 1,3–glucan and α − 1,6–glucan are given in Figure 5b and Figure 5c, respectively. Comparison of these results suggests that the glucan in the biofilm is composed of both α − 1,6–glucan and α − 1,3–glucan. Then, the ratio of the integrals of the C1 peak was calculated to estimate the component ratio of α − 1,6–glucan to α − 1,3–glucan in the biofilm. The ratio of the area subtended by the α − 1,3–glucan -derived peak to that of the α − 1,6–glucan-derived one was 1:0.27. Therefore, the glucan in the biofilm of the *S. mutans* culture is proved to be α − 1,3-rich.

An additional characterization with Wide-Angle X-ray Diffraction (WAXD) (shown in Appendix A) confirmed the presence of a significant fraction of α − 1,3–glucans in the biofilm of the *S. mutans* culture. WAXD data also revealed a lower crystallinity for biofilm α − 1,3–glucans as compared to a synthetic α − 1,3–glucan sample [30]. However, WAXD also confirmed that a conspicuous crystalline fraction resisted Sodium Dodecyl Sulfate (SDS) treatment.

Figure 6a shows a Raman map of the *S. mutans* biofilm along the in-depth *z*-axis. The thickness of the biofilm was ~22 μm. The map, which was non-destructively taken in confocal mode with a spatial resolution of 670 nm and focal plane displacements of ~100 nm, revealed the internal structure of the biofilm. In (b), (c), and (d), average Raman spectra are shown in the interval 400~600 cm^−1^ as collected along the line at *z* = 0, 10, and 22 μm, respectively. Similarly, average spectra in the interval 1400~1600 cm^−1^ are shown in (e), (f), and (g), respectively. From the intensity ratios, *R*^(1)^_α3/α6_ = *I*_520_/*I*_555_ and *R*^(3)^_α3/α6_ = *I*_1568_/*I*_1482_, the fractions of α − 1,3–glucans, F_α3_, were calculated according to calibration plots in Figure 3a,c, respectively. Along the biofilm *z*-axis, the fractions of α − 1,3–glucans were clearly more abundant in both the outermost layer (*z* = 0~5 μm; >90%) and in the zone of contact with the substrate (*z* = 15~22 μm; 50~60%). Conversely, in the central zone of the biofilm (*z* = 5~15 μm), F_α3_ values were the lowest (~3% and 15% according to *R*^(1)^_α3/α6_ and *R*^(3)^_α3/α6_, respectively). The agreement between the two Raman methods of F*_α_*_3_ measurement was good (cf. values in inset) except for the middle zone. An explanation for this discrepancy could reside in the overlap of C=C ring signals from different molecules. For example, tryptophan possesses a strong signal from ring C=C stretching at around 1560 cm^−1^ [36]. Tryptophan is contained in cells proteins and also in the glucansucrase enzyme produced by *S. mutans* to synthesize α − 1,3–glucans [37]. Therefore, its contribution to the 1568 cm^−1^ band could be more prominent in the middle zone of the biofilm, which, unlike external biofilm zones, is congested with *S. mutans* cells.

The layered architecture of the biofilm, as proved by confocal Raman spectroscopy, is clearly linked to the need of a complete isolation with respect to infiltration of foreign molecules from the environment. *S. mutans* is indeed known to possess the unique ability to convert sucrose into extracellular insoluble glucans, which support bacterial adhesion/cohesion forming the exopolysaccharide matrix [38]. In the oral cavity, such a matrix enables localized acidification within biofilms by impeding buffering by saliva and alkaline environment while allowing diffusion of dietary sugars that provide new “energy” to rejuvenate the biofilm environment [39,40,41]. The present in-depth mapping data by high-resolution confocal Raman spectroscopy fit with the generally accepted model of exopolysaccharide envelope that modifies the diffusion properties of the biofilm, thus creating and preserving an acidic microenvironment.

### 2.3. Raman Markers of Cocultures’ Biofilm Exopolysaccharides

Figure 7 shows Raman spectra collected in the wavenumber interval 600~1200 cm^−1^ for *S. mutans* and *S. sanguinis* individual cultures and cocultures at 24 h (cf. labels in inset). In this spectral zone, Raman markers for α– and *β*–glucans can be found at 948 and 892 cm^−1^, respectively. The ratio between these two Raman signals, *R_α/β_*, thus links to the volumetric fraction of *α*– to *β*–glucans [42,43]. The *R_α/β_* ratios computed for the set of culture/coculture investigated are given in inset to each spectrum in Figure 7a–e and comparatively plotted in Figure 7f with related statistics (cf. labels in inset).

This latter plot shows a gradual decreasing in α–glucan fraction with decreasing *S. mutans* population in coculture, reaching a minimum value for the culture of *S. sanguinis* alone (~50% that of the culture of *S. mutans* alone). This difference could be attributed to a decreasing trend for the fraction of *α* − 1,3–glucans, which can only be synthesized by *S. mutans* [38]. According to the calibration plot in Figure 3b, the ratio *R*^(2)^_α3/α6_ = *I*_948_/*I*_919_ could, in principle, be used to quantitatively evaluate the fraction of *α* − 1,3–glucans, *F_α_*_3_. In the spectra of Figure 7a–e, the signal at 919 cm^−1^ appears as an intermediate band between those at 948 and 892 cm^−1^. However, this approach, which was successfully used in previous assessments of biofilms in *Candida* species [26,31], cannot be applied to biofilm produced by dental bacteria because, unlike the case of culturing *Candida* species, sucrose is externally added to bacterial cultures of streptococci (cf. forthcoming Section 4.1). Sucrose molecules present strong Raman signals in the C–C stretching region at 851, 872, and 922 cm^−1^, this latter band strongly overlaps the *α* − 1,6–glucan fingerprint at 919 cm^−1^ (cf. sucrose-related bands labeled with sharps in Figure 7a–e). The impossibility to deconvolute the contributions from *α* − 1,6–glucans and externally added sucrose to this signal impedes the application of the *R*^(2)^_α3/α6_ = *I*_948_/*I*_919_ Raman algorithm.

High spectrally resolved Raman spectra in Zone I (400~600 cm^−1^) are shown in Figure 8a–e for cultures and cocultures (at 24 h) with *S. mutans* to *S. sanguinis* ratios of 1:0, 1:1, 1:4, 1:8, and 0:1, respectively. The trend of the two C–C–OH deformation bands, appearing at 555 and 520 cm^−1^ (cf. vibrational modes and Raman intensity ratio, *R*^(1)^_α3/α6_ = *I*_520_/*I*_555_, in Figure 1a and Figure 2a, respectively), was used to quantitatively determine the fraction, F*_α_*_3_, of *α* − 1,3–glucans contained in the biofilms according to the master plot shown in Figure 3a. The obtained F_α3_ data are displayed in the plot in Figure 3a and replotted in Figure 8f with their statistical significance. As seen, the presence of *S. sanguinis* in coculture at any fraction ≥50% reduced the content of *α* − 1,3–glucans in the biofilms to less than a half (i.e., from ~87% to ~38%) with no statistical significance among different cultures except for comparisons with the culture of *S. mutans* only. There were also additionally important details by which the spectra of different cocultures differed in the 400~600 cm^−1^ spectral interval. They can be summarized in the following three items (cf. Figure 8a–e and band assignments in Appendix A): (i) Two additional bands were found at 531 and 572 cm^−1^, whose relative intensity increased with increasing initial *S. sanguinis* population; (ii) a clear signal at ~478 cm^−1^ was detected only in the Raman spectrum of the 1:1 coculture (labeled with a blue sharp in Figure 8b); and (iii) a relatively strong signal was detected at ~493 cm^−1^ only in the culture of *S. sanguinis* alone (labeled with a green sharp in Figure 8e). Regarding the above item (i), the mentioned two bands correspond to C2–C3–O(–C1) (531 cm^−1^) and C5–C6–O(–H) (572 cm^−1^) bond deformation, and are both expected in *α* − 1,3–glucans. Their enhancement could be ascribed to the formation of shortened oligomers, to the presence of glucofuran rings, and/or to C–C–C bending or C–O torsion in glycogen (*α* − 1,4 glycosidic bonding) [44]. This point will be further considered in the forthcoming discussion section. The above item (ii), which is a signal only observed in the 1:1 coculture (cf. Figure 8b), corresponds to a wavenumber characteristic of S–S bond stretching in a ring structure [45].

This molecular feature is not intrinsic to the biofilm polysaccharide structure and its origin should be traced back to the competitive strategy of *S. mutans*, as discussed in the forthcoming Section 3.2. Finally, regarding the above item (iii), the 490 cm^−1^ signal, peculiar to the *S. sanguinis* culture, is characteristic of C–C–C bending in glycogen molecules [44,46,47]. When cultured in the presence of oxygen, which is the case here, *S. sanguinis* is known to accumulate large amounts of intracellular polysaccharide with prevalence of glycogen molecules [48]. Compared with other *Streptococcus* species, *S. sanguinis* exhibited the highest rate of glycogen synthesis and intracellular storage [49]. Moreover, under conditions of carbohydrate starvation, *S. sanguinis* was found to strongly prevail above *S. mutans* at least partly because of the intracellular possession of glycogen as energy source [50]. Upon matching the microscopic observation in Figure 4e with the Raman data in Figure 8e, it is suggested that, under the present culture conditions, *S. sanguinis* preferred to accumulate intracellular glycogen in addition to spending energy in building up a biofilm structure.

An additional spectral zone of interest with respect to the metabolism of bacteria in a coculture could be located in the wavenumber interval 1600~1800 cm^−1^. High spectrally resolved Raman assessments in this wavenumber zone are shown in Figure 9a–e for cultures/cocultures at 24 h with *S. mutans* to *S. sanguinis* ratios of 1:0, 1:1, 1:4, 1:8, and 0:1, respectively. This spectral zone, which is dominated by stretching vibrations of the carbonyl C=O bonds, presented four distinct bands at ~1723, 1735, 1763, and 1786 cm^−1^ (cf. band assignments as summarized in Appendix A).

The presence of several different wavenumbers for the C=O bands indicates a variety of locations in which the carbonyl unit has formed as a result of environmentally assisted biofilm processes. Note, however, that some of the above band components displayed a quite weak intensity, comparable with the noise level. Therefore, it is difficult to use these bands to draw final conclusions on their specific meaning. An important hint for the interpretation of C=O stretching signals is the observation that their intensity, comparatively weak in single-cultured bacteria (cf. Figure 9a,e), becomes gradually enhanced with increasing initial fraction of *S. sanguinis* in coculture (cf. Figure 9b–d). This trend might suggest that the gradual enhancement of carbonyl groups is associated to the competitive strategy adopted by both *S. sanguinis* (i.e., its ability to produce hydrogen peroxide) [11]. This point is further discussed in the forthcoming Section 3.2.

Phillips et al. [51] developed calibration plots using the ratio between the 1730 and the 941 cm^−1^ bands for a quantitative determination of the degree of substitution of carbonyl units, *S*, in a number of different carbohydrate structures. These researchers found that all plots were linear with nearly the same slope. Using that calibration slope for the intensity ratio of the 1725 to the 949 cm^−1^ bands in the present spectra, one could compute an increasing amount of carbonyls with increasing fraction of *S. sanguinis* in coculture up to more than 50% (cf. Figure 9f).

### 2.4. Raman Imaging of Water-Insoluble Exopolysaccharides Fractions

In order to spatially visualize the locations of *α* − 1,3–glucan metabolites within culture/cocultures of *S. mutans* and *S. sanguinis*, Raman images were collected with sub-micrometric spatial resolution (Figure 10a–e; cf. labels in inset). This procedure made it available a large number of spectra (in the order of 10^6^ per each map), which also served to confirm the statistical validity of trends recorded on average spectra (i.e., as described in the previous sections). In this perspective, Raman imaging could be considered as complementary to the Raman approach based on averaging over spectra collected with a relatively low-magnification optical probe (20×) (i.e., Figure 8). Figure 10f shows average values of *α* − 1,3–glucan fractions, F_α3_, as computed by averaging *R*^(1)^_α3/α6_ = *I*_520_/*I*_555_ ratios over the entire Raman images (*n* = 3 for each culture/coculture), according to the calibration plot in Figure 3a. F*_α_*_3_ values from the two different data-collection procedures (i.e., average spectra on large areas in Figure 8f vs. average of a large number of spectra from images in Figure 10f) showed agreement within ±10%, thus statistically validating the obtained data. 

In addition to strengthening statistical validation, the Raman visualization of *α* − 1,3–glucans’ distribution in live biofilms provided us with vivid snapshots of the competition between the two antagonist streptococci. The morphology of *α* − 1,3–glucan distribution in the *S. mutans* culture was that of a dense and continuous “blanket” (Figure 10a), while no such Raman signals could be observed in the *S. sanguinis* culture (Figure 10e). Even amounts of streptococci (i.e., the 1:1 coculture) led to a continuous but conspicuously less dense *α* − 1,3–glucan distribution (Figure 10b) while higher fractions of *S. sanguinis* (i.e., 1:4 and 1:8) discontinued the *α* − 1,3–glucan “blanket”, but yet they displayed regions of its high density (Figure 10c,d, respectively). The spatial architecture of the biofilm formed in *S. mutans*/*S. sanguinis* cultures/cocultures is an important feature since it visualizes biofilm impermeability. This point will be discussed in detail in the next section.

## 3. Discussion

### 3.1. Biofilm Vibrational Markers and Competition/Coexistence Strategies

In a spectroscopic probe of competitive mechanisms between different streptococcal species coexisting in the same ecological niche, we confirmed here that *S. mutans* and *S. sanguinis* engage in reciprocal antagonistic interactions to gain spatial supremacy and documented new details of their interactions. Coexistence of the concurrently inoculated streptococci in various fractions involved fundamental alterations of the exopolysaccharide glucans’ structure of the formed biofilm. Data from Figure 7 demonstrated a gradual reduction (down to ~50%) in the *α*– to *β*–glucans ratio, *R_α/β_*, with the volume fraction of water-insoluble *α* − 1,3–glucan molecules also being reduced by ~50% for any fraction of *S. sanguinis* added in coculture (cf. Figure 8 and Figure 10). Raman imaging then confirmed that only *S. mutans* creates abundant *α* − 1,3–glucan molecules in the biofilm and unveiled the spatial dependence of biofilm structure on *S. sanguinis* fraction (Figure 10). 

The percentage of *α* − 1,3–glucans in biofilm reduced to more than half at any fraction of *S. sanguinis* ≥50% in coculture, but spatial homogeneity in the biofilm structure was only preserved for the 50% fraction (cf. Figure 10b). Concurrently, oxidation at various sites in the glucans’ molecular chain was detected, as proved by a series of C=O carbonyl stretching bands located in the narrow spectral region 1720~1790 cm^−1^ (Figure 9). The origination of short oligomers and their subsequent oxidation can cause the formation mid-chain glucuronic acids (at C6 position), carbonyls at C2 to form oxo–glucose units, as well as carbonyls at the non-reducing end from lytic C4 oxidation (glucono–D–lactone units) [52]. However, breakage of glucan chains and subsequent formation of carbonyl bonds at various molecular sites necessarily involve interactions with oxygen radicals to form shorter oligomers as a consequence of glycosydic bond cleavage [52,53]. It should be noted that contributions to the detected esters from lipid metabolites from bacteria could not be completely ruled out. Nevertheless, the observed progressive increase in such signals in cocultures increasingly rich in *S. sanguinis* (cf. Figure 9), while significantly weaker in both *S. mutans* and *S. sanguinis* single cultures, supports the hypothesis that the main contribution to ester bands is a consequence of the interaction between the two bacterial species.

Possible mechanisms for the action of oxygen radicals are discussed later in this section. Figure 11 shows a draft of possibly oxidized carbonyl sites (compiled according to Ref. [52]) as positioned in a glucan chain cleaved by HO• radicals; the respective (guessed) Raman frequencies are given in inset. The draft also envisages the possible formation of transient C=C bonds, which could contribute the intensity of the observed ~1560 cm^−1^ band. Indeed, the *R*^(3)^_α3/α6_ = *I*_1568_/*I*_1482_ ratio increased from 0.08 to 0.13 with increasing *S. sanguinis* fraction in coculture (cf. Appendix A). According to the calibration plot in Figure 3c, such an increase apparently indicates a nearly doubled *α* − 1,3–glucan fraction, F_α3_ (from ~20 to 40%), while one would expect a decrease in the *α* − 1,3–glucan metabolite with decreasing fraction of *S. mutans* in coculture (cf. Figure 8f and Figure 10f). This trend confirms that the C=C signal at 1560 cm^−1^ is contributed by different molecules; not only the transient oxidation products of biofilm *α* − 1,3–glucans (and related glucansucrase enzymatic molecules) as produced by *S. mutans*, but also cells proteins (i.e., tryptophan ring vibrations) present in any of the two bacteria. Note that this circumstance rules out the possibility to use the *R*^(3)^_α3/α6_ Raman ratio for calibrating the *α* − 1,3–glucan fraction in live biofilm, but yet allows reasonably estimating *α* − 1,3–glucan fractions at specific locations lacking bacterial cells (e.g., upper and lower layers of biofilm cross sections; cf. Figure 6).

While micrographs in Figure 4 showed a thick and continuous biofilm even in the culture initially inoculated with a quite minor fraction of *S. mutans* (i.e., 1:8; cf. Figure 4d), Raman data suggest that the formed biofilms were quite altered with respect to the structure of their exopolysaccharides for both 1:4 and 1:8 inoculation fractions of *S. mutans*. Regarding the three-dimensional structure of *S. mutans*’ biofilm (Figure 6a), in-depth confocal Raman map revealed strong gradients in glucan structure, with top and bottom layers ~1/3 of the biofilm thickness containing high fractions of *α* − 1,3–glucans (i.e., >90% and 50~60%, respectively). The conspicuous elimination of *α* − 1,3–glucan molecules, as observed in coculture including *S. sanguinis*, should render the biofilm much less impermeable to the external environment (a situation preferred by *S. sanguinis*).

This is evident in the α − 1,3–glucan-free domain-like structure at prevalent fractions of *S. sanguinis* (cf. Figure 10c,d) while the uniform reduction in α − 1,3–glucan (and biofilm impermeability) at 1:1 initial fraction could be the result of a yet somewhat “chaotic” bacterial distribution within the biofilm (cf. Figure 10b).

The origin of a consistent reduction in *α* − 1,3–glucan content with increasing fraction of *S. sanguinis* in coculture agrees with the linkage analyses by Kopec et al. [32], which showed that glucans in *S. sanguinis* biofilm are predominantly α − 1,6 –linked and have only a small amount of *α* − 1,3–linked glucose. However, the observed trend could also be contributed by the capacity of this species to produce H_2_O_2_ [11]. The hypothesis of a “radical-chemistry effect” is supported by the finding of Raman fingerprints for oxidation of shortened oligomers (cf. Figure 9). As mentioned above, cleavage of glycosidic bonds can only occur by the effect of HO• radicals in a reaction system of hydrolysis (glycoside hydrolase) [52,53,54]. Hydrogen peroxide is well known to possess a low but stable reactivity capable of generating reactive oxygen radicals that efficiently degrade polysaccharides [53]. It is also established, through accurate genomic analyses, that, unlike *S. mutans*, *S. sanguinis* can only produce one type of glucosyltransferase, which synthesizes soluble glucans [55,56,57,58]. This, together with the low attitude of *S. sanguinis* to form biofilm in BHI medium [33], explains why a culture of only *S. sanguinis* showed a negligible amount of biofilm (cf. Figure 4e). However, we also observed large amounts of biofilm in cocultures with predominant fractions of *S. sanguinis*, which confirms that this bacterium is capable of exploiting metabolites produced by *S. mutans*. Zhu and Kreth [59] have described in detail the antagonistic effect of H_2_O_2_ on the oral biofilm developed by *S. sanguinis*. Although the main effort of *S. sanguinis* is that of eliminating competitors with the goal of promoting the integration of H_2_O_2_-compatible species in the biofilm, the production of H_2_O_2_ also causes the release of extracellular DNA into the environment, which is used to promote biofilm formation (as it is observed here) and to support adaptation processes. *S. sanguinis* is known to produce alpha-hemolysis on blood agar; a characteristic linked to the ability of streptococci to oxidize hemoglobin in erythrocytes by secretion of oxygen peroxide [60]. *S. sanguinis* genes related to the generation of H_2_O_2_ are essential in repressing *S. mutans* [61,62]. Conversely, *S. mutans*, in order to protect itself against H_2_O_2_ damage, has genetically built up an efficient H_2_O_2_-resistance system [63,64]. Superoxide dismutase and eukaryotic-type serine/threonine protein kinase also boost up the ability of *S. mutans* to deal with damages from reactive oxygen species in order to coexist with *S. sanguinis* [65]. This chemical machinery is essential for the survival of *S. mutans* in healthy oral flora, in which *S. sanguinis* prevails. The present data indeed confirm this understanding by showing that *S. mutans* is capable to survive even under conditions of striking minority (cf. Figure 10d).

As mentioned above, independently existing unpaired electrons from reactive hydroxyl radicals, HO•, are required for the occurrence of glycosidic bond-cleavage [52,53,54]. In human oral environment, HO• formation from the non-radical H_2_O_2_ molecules generated by *S. sanguinis* entails the occurrence of a catalytic cycle triggered by two concurrent chemical circumstances: (i) the pro-oxidant activity of a reducing agent (e.g., ascorbic acid directly produced by bacteria or from the diet), and (ii) the reduction of intrinsic iron by superoxide ions O_2_•^−^ to produce O_2_ (by auto-oxidation) and Fe^2+^ ions. These latter ions then react with H_2_O_2_ to produce HO• according to Fenton reaction [53,61]. In the present in vitro experiments, environmental pH in the cocultures was set at homeostatic value (pH = 7.2 ± 0.2); iron ions were present in the BHI broth (16.85 μM) [66] and most likely were effectively sequestered by *S. sanguinis* [67]. The above metabolic machineries enabled *S. sanguinis* to open a significant amount of permeable domains in the biofilm built by *S. mutans* by both disrupting the α − 1,3–glucan structure and scavenging extracellular DNA to promote its own biofilm formation. These actions supported penetration of environmental water and newly dissolved O_2_ molecules, which trigger the formation of new radicals, promoted further hydrolysis and additional domain opening, and ultimately allowed *S. sanguinis* to proliferate.

### 3.2. “Chemical Weapons” in Active and Passive Competitive Exclusion

Genetic and biochemical analyses by Kreth et al. [11] demonstrated that diffusible substances produced by *S. mutans* and *S. sanguinis* (more specifically, mutacin and H_2_O_2_, respectively) were responsible for the antagonistic interactions ultimately leading to the observed mechanism of competitive exclusion. In that study, competition assays performed on biofilms showed that mutual exclusion by such inhibitory substances only occurred when the two species were inoculated in sequence (with the early inoculated one prevailing). Conversely, no competition occurred when both species were inoculated at the same time, thus leading to coexistence. This indeed was also the case of the present experiments, in which concurrent inoculation always led to coexistence whatever the fraction of *S. sanguinis* up to 87.5% (cf. biofilm micrographs in Figure 4). However, Raman imaging (Figure 10) revealed new details of the “chemical” competition strategy developed by *S. sanguinis* according to the mechanisms described in the previous section.

Since a bacterial strain is considered to be competitive if it causes a fitness decrease in a competitor strain [68], the present study confirmed that both the investigated streptococcal bacteria presented competitive characteristics. However, the competitive action by *S. sanguinis* resulted to be passive in nature since the production of inhibitory substances (oxygen radicals) only indirectly created environmental disadvantage to the competing species (i.e., rather than directly inducing rival cell damage or lysis). Classifications in the frame of passive competition usually foresee several different strategies. Bacteria might secrete digestive enzymes [69,70] or siderophores [71,72] to restrict competitor’s access to nutrients or alter metabolic regulation to enhance their own nutrient utilization [72]. In alternative, some strains have been reported to reduce the expression of costly genes upon exploiting expression products from other strains (i.e., the so-called “cheating effect”) [69,71,73,74]. Finally, passively competitive strains might produce specific molecules that impact space structure and enable space gain (e.g., surfactants or rhamnolipids) [75,76,77,78,79]. This latter was definitely the case of *S. sanguinis* in the present study. As first demonstrated here, this bacterium acted to locally discontinue the network of water-insoluble biofilm exopolysaccharides built by *S. mutans* in order to create niches of permeable biofilm in which it could more likely dominate.

Regarding the competitive strategy of *S. mutans*, in situ Raman experiments confirmed the secretion of polycyclic peptides containing thioether amino acids (cf. increase in the intensity of S–S band in the low-frequency spectrum of Figure 8b), but only in the 1:1 coculture. Unlike the case of *S. sanguinis*, the nature and the effect(s) of such competitive strategy could not be clearly visualized by Raman imaging. However, since lantibiotics, when synthesized by ribosomes and post-translationally modified to their biologically active forms, generally possess antimicrobial properties, the competitive strategy of *S. mutans* should be considered as active in nature [80]. A number of differently structured mutacins (i.e., classified as lantibiotic and nonlantibiotic structures) are active against at least 11 species of streptococci by docking receptor on the cell membrane, forming disrupting pores on it, and/or directly interfering with cell wall biosynthesis [81]. Lantibiotic mutacins, which possess a wider spectrum of activity than nonlantibiotic, contain post translationally modified peptides, including either lanthionine or methyllanthionine ring structures in addition to dehydrated amino acids [82]. According to Merritt and Qi [80], the bactericidal activity of specific mutacin lantibiotics (i.e., of Type A) features lipids as docking molecules for simultaneous pore formation and inhibition of cell wall synthesis. Conversely, Hossain and Biswas [83] reported about two specific nonlantibiotic mutacins (mutacins IV and V) produced by *S. mutans*, which synergistically act to inhibit the growth of other bacteria. *S. sanguinis* attempts to reduce mutacin production by inactivating the *S. mutans* competence-stimulating peptide, a quorum sensing signal inducing mutacin gene expression [84]. Nonlantibiotics (most active against closely related species) are unmodified peptides including one or two separate molecules [85,86], which accumulate intracellularly and interact with membrane receptors leading to competitors’ lysis. The present analyses of Raman spectra did not enable an exact discrimination of the lantibiotic or nonlantibiotic nature, neither to draw the chemical structure of the mutacins active against *S. sanguinis*. However, the signal from a sulfane–sulfur bond detected ~478 cm^−1^ (only in the 1:1 coculture; cf. blue sharp in Figure 8b) hinted to an S–S bond stretching in ring structures (i.e., present in lantibiotic molecules as lanthionine) while the short polymeric sulfur chains of nonlantibiotics are expected to produce a broader signal at ~10 cm^−1^ lower wavenumbers (not observed here given the perfectly symmetric morphology of the 478 cm^−1^ signal; cf. Figure 8b) [87,88,89,90]. This spectroscopic characteristic seems to confirm that *S. mutans* produces mutacins to kill peroxigenic streptococci [91]. Despite a lack of specificity for the S–S Raman signal, the availability of a Raman fingerprint for mutacins could be key in directly monitoring in situ and in time lapse of the level of *S. mutans* competitive activity in the presence of other bacterial species. This point deserves a more specific experimental approach in future studies. Provided that vibrational aspects of the chemical mechanisms behind the competitive behavior of *S. mutans* strains could further be clarified, novel functions could be discovered and a potential diagnostic approach could become available for and real-time on-site assessments of oral biofilm pathogenicity.

In summarizing the present findings, Figure 12a,b schematically depict biofilm zones in which *S. mutans* and *S. sanguinis* populations prevail, respectively. In the former case, which foresees a biofilm rich in water-insoluble (and impermeable) exopolysaccharides, *S. mutans* uses mutacins in the attempt to directly exterminate the *S. sanguinis* competitor according to an active competition strategy. In the latter case, *S. sanguinis* adopts a passive competition strategy and eliminates water-insoluble exopolysaccharides by producing oxygen radicals that open water-permeable “holes” in the biofilm structure. In these oxygen-rich zones, the *S. mutans* competitor could hardly dominate. In summary, Raman spectroscopy enabled us to retrieve and image so far unknown molecular-scale details about the sophisticated interspecies interactions that balance competition/coexistence within *S. mutans*/*S. sanguinis* microbial communities.

### 3.3. Raman Technology in Oral Health and Limitations of the Present Study

Oral diseases represent a major global public health issue with dental caries covering a prevalent economic burden and being responsible for a significant degradation of the patients’ quality of life [92]. Control of composition and metabolism of the oral flora is key in preventing oral diseases, but these factors can hardly be monitored with sufficient precision within the limited time of a clinical session. Raman spectroscopy has the potential to become a unique chairside technology to sensitively and rapidly detect changes at the molecular level both in tissue and oral flora. However, quantitative Raman analyses require the construction of detailed and comprehensive algorithms, which in turn necessitate the construction of complex biochemical and vibrational database.

In previous Raman studies of enamel and dentine tissue demineralization [93,94,95,96], imaging algorithms for estimating the risk of caries have been proposed, which utilize the totally symmetric stretching mode of phosphorous tetrahedra as a sensor of calcium vacancy concentration. Raman signals arising from the internal vibrations of the phosphate ion (PO_4_^3−^) are quite strong and dominate the experimental spectrum of enamel, which makes this Raman procedure so far the closest to an actual clinical use. Unlike the relatively straightforward algorithms for Raman analyses of demineralization in hard tissues, Raman analyses of pathogenic flora inhabiting the oral cavity require massive vibrational information from elementary organic molecules and precise interpretations of pronounced band overlap [26]. In order to reverse the spectral features into useful biochemical information at the molecular level in such complex systems, bioinformatics, statistics, and machine-learning algorithms should be constructed that enable physically sound spectral deconvolutions empowering the Raman method. A widely adopted method to assist Raman diagnosis and prognostic assessments of pathogenic oral flora relies on principal component analyses (PCA)-based chemometrics [23,97,98,99,100]. In case of speciation deficiencies by the PCA method, Raman barcoding has been proposed, which allows obtaining a greater depth in capturing structural details than merely comparing spectral morphologies as done in PCA analyses [26,27,29].

The present Raman analysis of streptococcal cocultures involved several molecules of difficult identification whose elementary spectra are yet conspicuously unknown (i.e., mutacin lantibiotics and nonlantibiotics). This circumstance made a complete spectral deconvolution by machine-learning algorithms (and related barcodes) of difficult application. Accordingly, only selected spectral areas could be analyzed. Specifically, we focused on distinct biomarkers that are associated with fingerprint bonds of key molecules: C–O–C glycosydic bonds in exopolysaccharides and S–S bonds in ring-structured mutacins. While an advantage of this partial approach is given by the straightforward data analysis and the relatively good discrimination efficiency, its main limitation resides in the fact that a wealth of important biochemical details yet remains buried in the unexplored spectral regions. The possibility of performing reliable real-time routine assessments of the stage of active competition of *S. mutans* in patients’ oral plaque would certainly represent a significant step forward for human clinical purposes. From this viewpoint, the Raman analyses presented here, although yet incomplete, demonstrate that signals from selected exopolysaccharides directly link to the state of antagonistic interactions between coexisting bacterial populations. This unique biochemical circumstance holds promise specifically for real-time quantitative analyses of oral plaque specimens and calls for further investigations on ex vivo samples.

## 4. Materials and Methods

### 4.1. Bacterial Strains and Culture Conditions

*S. sanguinis* strain ATCC10556 (serotype c) and *S. mutans* strain ATCC 700610 were purchased from American Type Culture Collection (ATCC; Manassas, VA, USA). Both strains were expanded in BHI broth (Nissui Pharmaceutical, Co. Ltd., Tokyo, Japan) at 37 °C and then cultured individually or at *S. sanguinis*-to-*S. mutans* concentration ratios of 1:1, 1:4, and 1:8 in 1.5 mL BHI broth (pH = 7.2 ± 0.2) with 0.5% sucrose (Nacalai Tesque, Inc., Kyoto, Japan) containing 0.5 mL human saliva (Normal Saliva, Single Human Donor; Lee Biosolutions, Maryland Heights, MO, USA) under anaerobic conditions (AnaeroPack^®^, Mitsubishi Gas Chemical Co., Tokyo, Japan). Human saliva samples were sterilized by membrane filtration through a sterile 0.22-μm filter (Dismic^®^ Membrane Filters, 0.22-μm pore size, Advantech Toyo Kaisha, Ltd., Tokyo, Japan). Cocultures were observed with a laser microscope (VK-x200 series, Keyence Co. Ltd., Osaka, Japan). 

Biofilms were washed in phosphate-buffered saline solution in order to remove the planktonic bacteria. The biofilms were then incubated in 1 mL of 0.5 M sodium hydroxide solutions (Nacalai Tesque, Inc., Kyoto, Japan) for 15 min and vortex-agitated for 15 s in order to detach the adhered biofilms from the plate and to dissolve water-insoluble glucans. The solution was then centrifuged at 5000 rpm for 10 min in order to separate the dissolved glucans from the bacteria embedded in the biofilms [101,102]. The amount of glucans was measured by phenol-H_2_SO_4_ (phenol-sulfuric acid method) at an absorbance of 490 nm and displayed as OD values. After this procedure, 200 µL of each of the resulting solutions was pipetted into separate wells of a 96-well flat-bottom microplate to measure the amount of glucans (Total Carbohydrate; mM) with the plate reader. To obtain a standard curve, 0, 0.063, 0.125, 0.25, 0.5, 1.0, 2.0, and 4.0 mM of carbohydrate standard was titrated using the Total Carbohydrate Assay Kit (Cell Biolabs, Inc., San Diego, CA, USA).

### 4.2. Nuclear Magnetic Resonance Analyses

Biofilms of *S. mutans* culture were washed in distilled water by centrifugation at 5000 rpm several times, then the biofilm precipitates were freeze dried. The reference α-1,3-glucan was synthesized by enzymatic polymerization as described in a previous paper [103]. In brief, recombinant GtfJ enzyme was incubated for two weeks at 30 °C in 50 mM citrate buffer (pH 5.5) containing 0.5 M sucrose and 0.01% NaN_3_. After incubation, the reaction mixture was centrifuged at 5000 rpm to separate insoluble product, then the precipitate was re-dispersed in distilled water followed by centrifugation several times, then freeze-dried. The reference α-1,6-glucan (Dextran 150,000) was purchased from FUJIFILM Wako Chemicals (Japan). The dried biofilms and reference α-glucans were dissolved in 2% NaOD in D_2_O at a concentration of 40 mg/mL containing 3.5 mg NaBD_4_, ^13^C NMR measurement were performed using a JNM-A500 FT-NMR system (500 MHz, JEOL Ltd., Tokyo, Japan).

### 4.3. Raman Library, In Situ Raman Spectroscopy, and Raman Imaging

A library of reference Raman spectra was built up, which contained more than 40 elementary compounds and including polysaccharides (chitin, β-1,3-glucans, β-1,6-glucans, α–1, 3–glucans, α–1, 6–glucans), mono- and disaccharides (trehalose, β-D-glucose, D-dextrose), anionic lipids (phosphatidylglycerol, phosphatidylethanolamine, cardiolipin), and other key metabolites such as adenine and other nucleic acids. The reference α-1,3-glucan samples were provided by Tokyo University [104]. The library was compiled with high spectrally resolved Raman spectra collected with a triple-monochromator device (T-64000; Jobin-Ivon/Horiba Group, Kyoto, Japan) equipped with a nitrogen-cooled charge-coupled device detector (CCD-3500V, Jobin-Ivon/Horiba Group, Kyoto, Japan). The excitation source was the 514 nm line of an Ar-ion laser operating with a nominal power of 20 mW. The spectral resolution was better than 1 cm^−1^.

High spectrally resolved Raman spectra were collected in situ on bacterial cocultures using a spectrometer specially designed for measurements on biological samples (LabRAM HR800, Horiba/Jobin-Yvon, Kyoto, Japan). An optical circuit set in confocal mode with a 20× objective lens and employing a holographic notch filter concurrently enabled high signal efficiency and high spectral resolution. The wavelength of the incoming light was a 532 nm emitted by a solid-state laser source operating at 10 mW. The Raman scattered light was monitored by means of a single monochromator interfaced with an air-cooled charge-coupled device (CCD) detector (Andor DV420-OE322; 1024 × 256 pixels). The acquisition time for a single spectrum was typically 10 s. A spectral resolution of better than 1 cm^−1^ was achieved by concurrently collecting (at each measurement) an internal reference signal from a selected neon lamp to calibrate the spectrometer. Twenty spectra collected at different locations (over a total area of ~2 mm^2^) were obtained for each bacterial culture/coculture and averaged in order to obtain representative spectra. All presented Raman spectra were recorded on samples collected after 24 h in culture under exactly the same conditions.

Raman imaging was recorded with a dedicated Raman device (RAMANtouch, Nanophoton Co., Minoo, Osaka, Japan) operating in microscopic measurement mode (100× lens; numerical aperture, NA = 0.9) with confocal imaging capability in two dimensions. This equipment was capable of achieving an in-plane spatial resolution up to 300 nm by exploiting a specially designed spectrograph with completely compensated aberration, which greatly improves spatial resolution of the optical objective. A dedicated confocal optics also enabled high spatial resolution (~670 nm) along the out-of-plane *z*-direction, as previously calibrated using a preliminary line-scan along the *z*-axis to find signals from a calibration substrate. As an additional peculiarity, this Raman microscope was capable to achieve ultra-fast simultaneous image acquisition of up to 400 spectra, which greatly reduced the laser irradiation time and rendered the Raman measurements compatible with life. The excitation source was a 532 nm solid-state laser with highly precise operation (TEM00 beam quality). The spectral resolution was ~2 cm^−1^ (spectral pixel resolution equal to 0.3 cm^−1^/pixel) with an accuracy in laser spot spatial location of 100 nm, as achieved by the special optical arrangement developed by Nanophoton Co. and peculiar to this instrument. Raman hyperspectral images were generated using commercially available software (Raman Viewer, Nanophoton Co., Minoo, Osaka, Japan). Raman images were generated using intensity ratios from normalized spectra. In order to minimize errors related to spectral resolution and possible shifts in band position, the average intensity of the pixels at the band nominal location ± 3 pixels was used instead of single-pixel intensity. Lateral displacement steps of 500 nm were adopted for the laser focal point in scanning selected areas of the samples.

### 4.4. Machine Learning Algorithm for Spectral Deconvolution

Experimental Raman spectra were subjected to polynomial baseline subtraction and deconvolution into a series of Gaussian-Lorentzian band components. The baseline subtraction procedure was performed using options available in commercial software (LabSpec 4.02, Horiba/Jobin-Yvon, Kyoto, Japan) with fixed criteria for all collected spectra. All spectra were analyzed after intensity normalization to the strongest signal in the collected spectral interval. Detailed descriptions of spectral deconvolution criteria have been reported in previously published papers [23,24]. An automatic solver exploiting a linear polynomial expression of Gaussian-Lorentzian functions was iteratively run to match average experimental spectra for minimum scatter (better than 95% confidence interval). Sub-bands for deconvolution were selected according to a computer program picking from spectra of pre-selected compounds belonging to the aforementioned library. Upon pre-selection of elementary molecules from the library, the algorithm located the best-fitting combination of the experimental spectra by preserving relative intensities, spectral positions, and full-width at half-maximum values from each elementary compound (i.e., within ±3 cm^−1^; a boundary value selected by considering the resolution of the spectrometer and the possibility of slight alterations in molecular structure). Those mathematical constraints allowed univocal deconvolution of the experimental spectra. An important difference with previous studies was that, in the case of biofilm analyses, the Raman library of elementary molecules missed several key molecules (i.e., mutacin lantibiotics and nonlantibiotics). Accordingly, only specific spectral regions could properly be deconvoluted and analyzed. 

### 4.5. Statistical Analysis

The statistical relevance of the experimental data was analyzed by computing mean values and standard deviations. Statistical validity was evaluated by applying the ANOVA analysis of variance [105]. Depending on sample, values *p* < 0.05 or <0.01 were considered as statistically significant and labeled with one and two asterisks, respectively.

## 5. Conclusions

In this study, we used a combined Raman approach of average spectra and high-resolution imaging to explore in vitro biofilm interactions between *S. sanguinis*, a benign bacterium supporting oral homeostasis and health, and *S. mutans*, a major pathogen causing human tooth decay. The Raman analyses clarified how the two species interacted by competing for space as a function of their relative population fractions in concurrently inoculated cocultures. Whatever the initial bacterial fraction in coculture, we always observed a continuous biofilm structure, never completely dismantled by *S. sanguinis*. However, Raman quantification of glucan metabolites in live biofilms indicated that *S. sanguinis* followed a passive path for securing space competition by releasing oxygen radicals in the biofilm environment. These radicals disrupted the network of water-insoluble *α* − 1,3 −glucans by glycosydic bond cleavage and subsequent oxidation of the shortened oligomers, thus yielding to “roof leakage” inside the biofilm from the external environment. On the other hand, *S. mutans* adopted an active competitive strategy by secreting mutacins to directly inhibit of *S. sanguinis* growth. Raman biomarkers for these two different strategies were: (i) the intensity ratio between the 520 and 570 cm^−1^ Raman bands (i.e., the only surviving approach of three different Raman proposed ones), which was quantitatively linked to the volumetric ratio between *α* − 1,3– and *α* − 1,6–glucans, and (ii) the 478 cm^−1^ signal from S–S bonds in ring lantibiotic structures.

In summary, this paper shows how Raman spectroscopy could be used in promptly identifying biofilm composition, in understanding molecular dynamics in the chaotic environment of oral biofilms, and in clarifying competitive bacterial interactions in the details of their complexity. Further developments of quantitative Raman algorithm and portable Raman instrumentation will provide researchers and medical practitioners with a unique tool to monitor in real time biofilm structures and bacterial metabolism in oral health and disease.

## Figures and Tables

**Figure 1 ijms-24-06694-f001:**
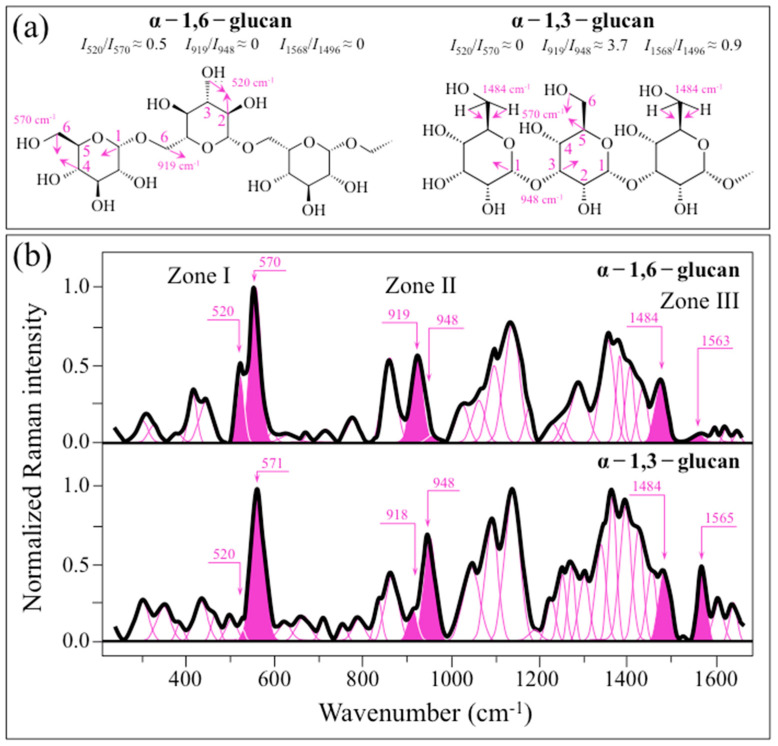
(**a**) Structures and fingerprint Raman vibrational modes of α − 1,6– and α − 1,3–glucans and (**b**) their respective spectra as collected on pure compounds are pictured; three spectral zones are specified in inset as Zones I, II, and III, which correspond to vibrational modes of C–C–O(H) deformation, C–O–C stretching, and H–C–OH bending, respectively. The band at 1563 cm^−1^ represents C=C ring bonds and arises from the residual presence of glucansucrase enzymatic molecules used in the preparation of the α − 1,3–glucan structure. The numbers in inset to the spectra represent peak positions and are given in cm^−1^.

**Figure 2 ijms-24-06694-f002:**
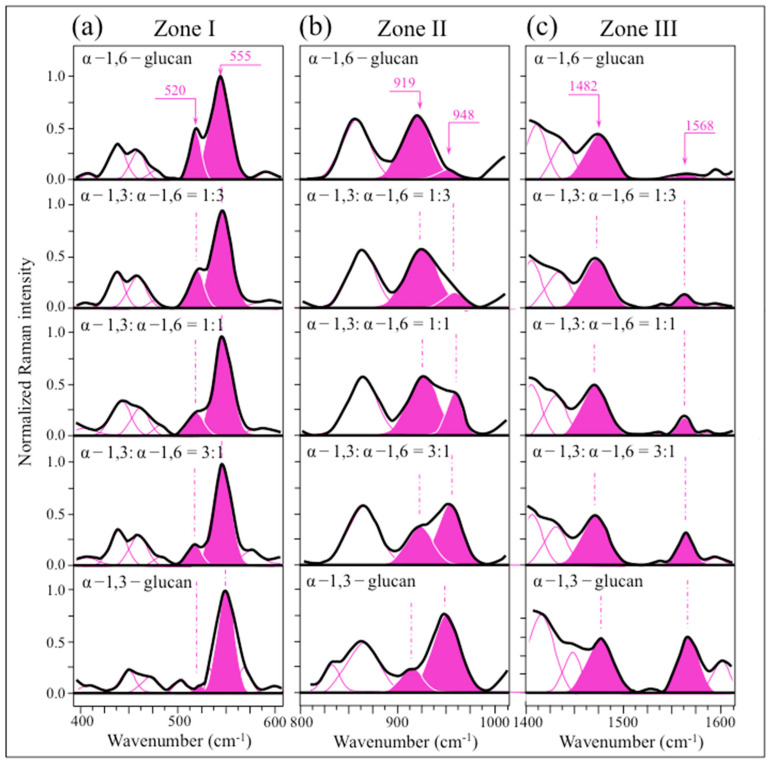
Raman spectra collected on mixtures of elementary compounds in different fractions of α − 1,3– to α − 1,6–glucans: 0:1, 1:3, 1:2, 3:1, and 1:0 (cf. labels) in spectral region (**a**) Zone I, (**b**) Zone II, and (**c**) Zone III. Wavenumber fingerprint Raman signals (in cm^−1^) in each zone are labeled in inset.

**Figure 3 ijms-24-06694-f003:**
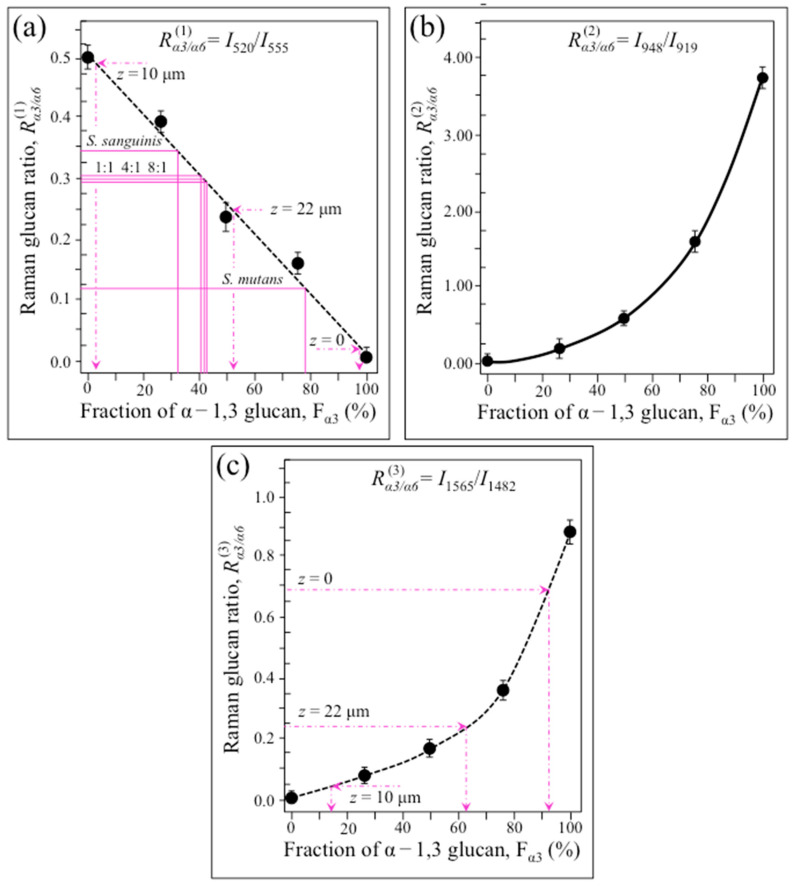
Dependence of Raman intensity ratios: (**a**) *R*^(1)^_α3/α6_ = *I*_520_/*I*_555_, (**b**) *R*^(2)^_α3/α6_ = *I*_948_/*I*_919_, and (**c**) *R*^(3)^_α3/α6_ = *I*_1568_/*I*_1482_ are plotted as a function of α − 1,3– to α − 1,6–glucans fraction, *F*_α3_. All plots could, in principle, be used to evaluate the glucan structure of live biofilm, but their suitability for this purpose differs, as discussed in the text. Scatters in the plot represent standard deviations.

**Figure 4 ijms-24-06694-f004:**
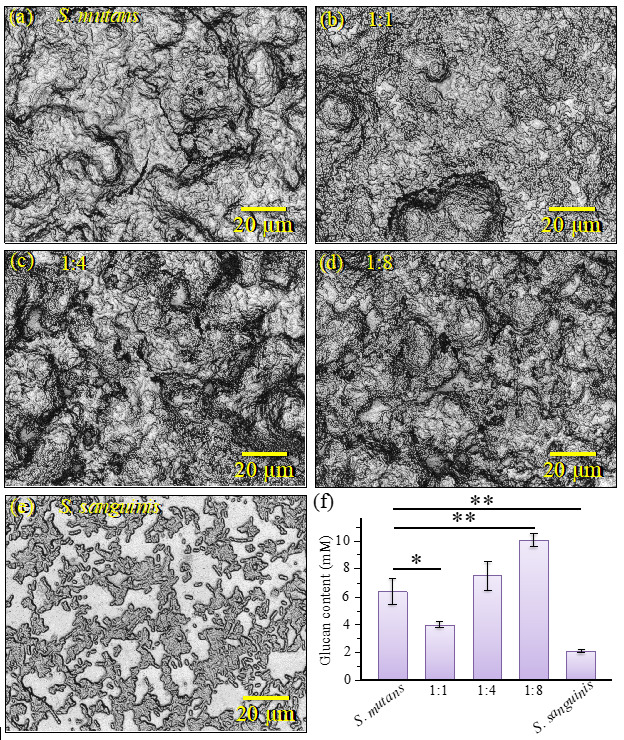
Optical micrographs of concurrently inoculated cocultures (at 24 h) with different *S. mutans*– to –*S. sanguinis* ratios: (**a**) 1:0, (**b**) 1:1, (**c**) 1:4, (**d**) 1:8, and (**e**) 0:1. Abundant biofilm formation is observed at any bacterial fraction except for the culture of *S. sanguinis* alone, which showed a negligible amount of biofilm. In (**f**), the amount of glucans is shown as collected on solutions of completely dissolved (soluble and insoluble) glucans (*n* = 3; * → *p* < 0.05; ** → *p* < 0.01).

**Figure 5 ijms-24-06694-f005:**
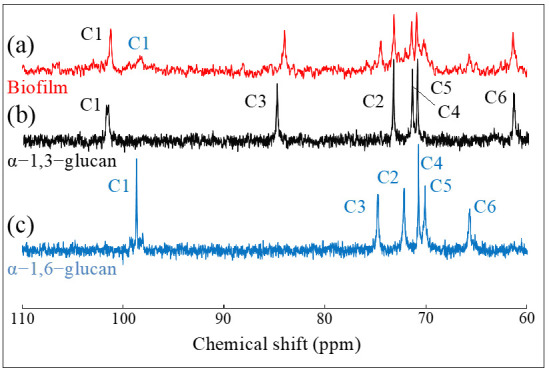
^13^C-NMR spectrum of biofilm in *S. mutans* culture (**a**), α − 1,3–glucans (**b**), and α − 1,6–glucan (**c**).

**Figure 6 ijms-24-06694-f006:**
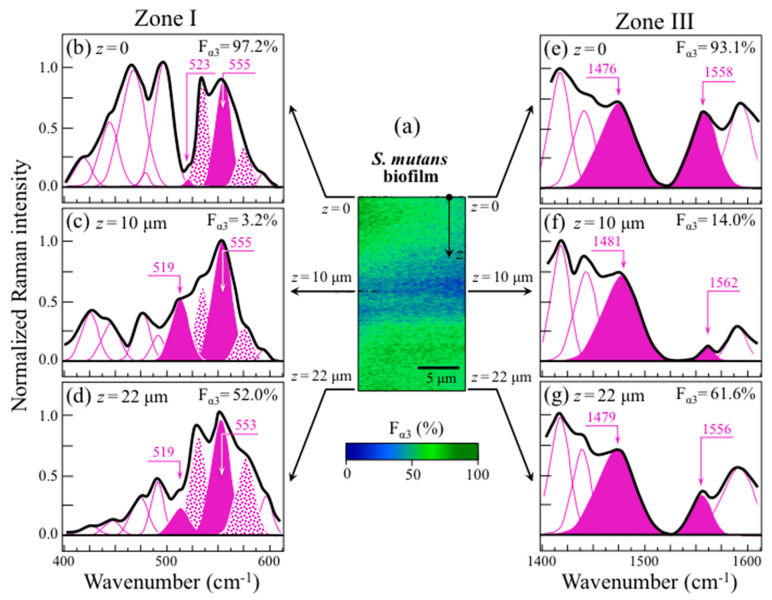
(**a**) Map of *S. mutans* biofilm along the in-depth *z*-axis as non-destructively collected by confocal Raman spectroscopy; average Raman spectra are shown for spectral Zone I (400~600 cm^−1^; (**b**–**d**)) and Zone III (1400~1600 cm^−1^; (**e**–**g**)) as line averages at in-depth abscissas *z* = 0, 10, and 22 μm, respectively (cf. labels in inset). Fractions of α − 1,3–glucans, F_α3_ (shown in inset to each spectrum), were calculated from the intensity ratios, *R*^(1)^_α3/α6_ = *I*_520_/*I*_555_ and *R*^(3)^_α3/α6_ = *I*_1568_/*I*_1482_, according to the calibration plots in Figure 3a and Figure 3c, respectively.

**Figure 7 ijms-24-06694-f007:**
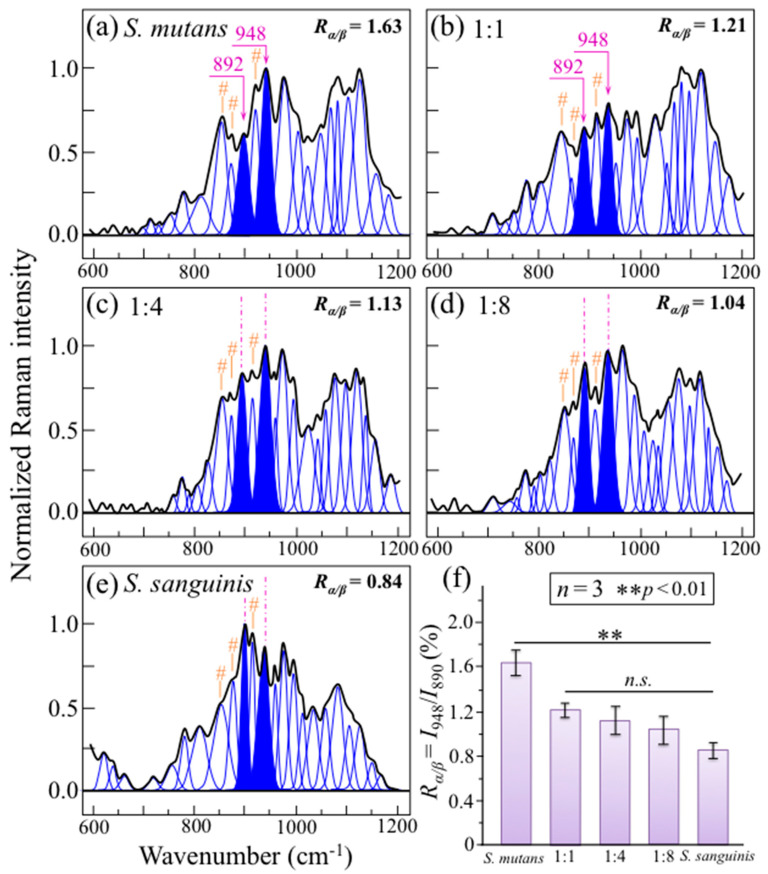
Raman spectra in the wavenumber interval 600~1200 cm^−1^ for *S. mutans* and *S. sanguinis* cultures/cocultures at 24 h (focal plane of the Raman probe on the sample surface): (**a**) 1:0, (**b**) 1:1, (**c**) 1:4, (**d**) 1:8, and (**e**) 0:1; the computed intensity ratios, *R*_α/β_, between Raman markers for *α*– and *β*–glucans (at 948 and 892 cm^−1^, respectively) are shown in inset to each figure and comparatively plotted in (**f**) with related statistics (cf. labels in inset). The signals at ~851, 972, and 919 cm^−1^ (labeled with orange sharps in (**a**–**e**)) are strongly contributed by signals from the sucrose externally added for culturing bacteria. *n.s.* = non significant.

**Figure 8 ijms-24-06694-f008:**
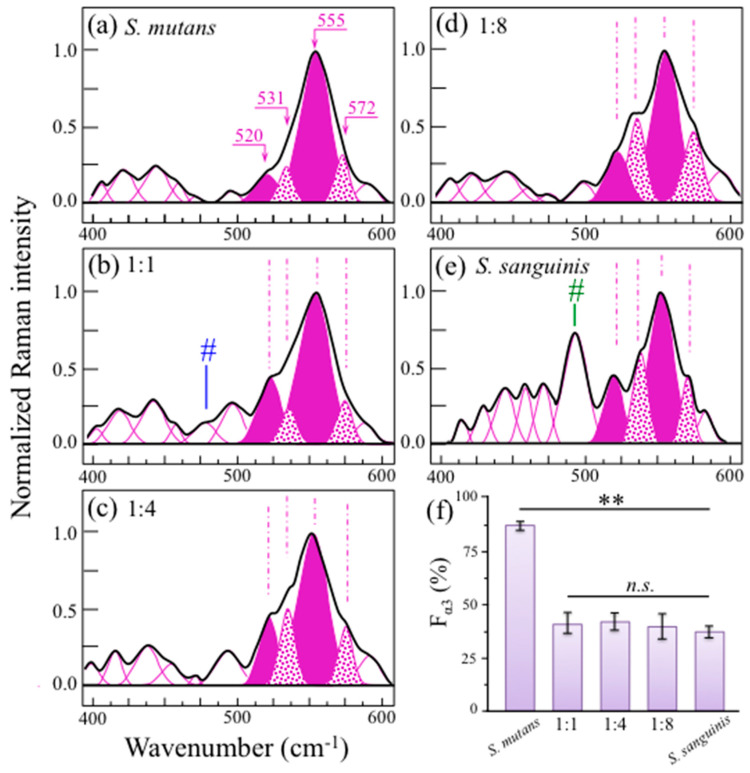
Raman spectra (probe focal plane on the sample surface) in Zone I (400~600 cm^−1^) as collected on cultures/cocultures (at 24 h) with *S. mutans* to *S. sanguinis* ratios of: (**a**) 1:0, (**b**) 1:1, (**c**) 1:4, (**d**) 1:8, and (**e**) 0:1; two C–C–OH deformation bands, appearing at 555 and 520 cm^−1^ (cf. labels) were used to compute the Raman intensity ratio, *R*^(1)^_α3/α6_ = *I*_520_/*I*_555_, from which the fractions, *F*_α3_, of *α* − 1,3–glucans contained in the biofilms were computed according to the plot in Figure 3a. Computed F_α3_ values are displayed in (**f**) together with their statistical significance (*n* = 3; ** → *p* < 0.01; *n.s.* = non-significant). Bond deformation bands at 531 cm^−1^ (C2–C3–O(–C1)) and 572 cm^−1^ (C5–C6–O(–H)) (cf. labels in inset) are only expected in *α* − 1,3–glucans. Bands labeled with blue and green sharps represent S–S bond stretching in ring structures and C–C–C bending in glycogen molecules, respectively.

**Figure 9 ijms-24-06694-f009:**
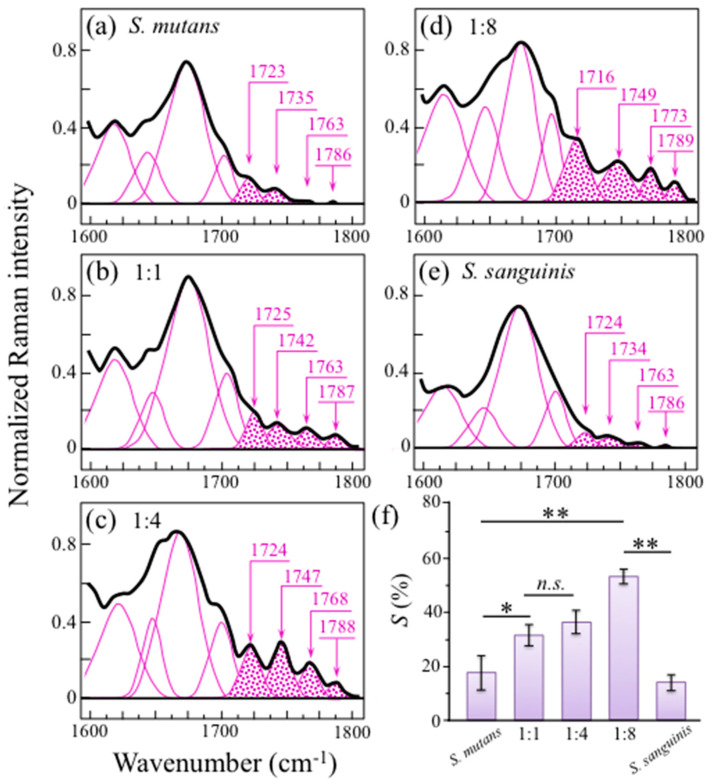
Raman spectra in the wavenumber zone 1600~1800 cm^−1^ as collected (probe focal plane on the sample surface) on cultures/cocultures (at 24 h) with *S. mutans* to *S. sanguinis* ratios of: (**a**) 1:0, (**b**) 1:1, (**c**) 1:4, (**d**) 1:8, and (**e**) 0:1; this spectral zone is dominated by stretching vibrations of the carbonyl C=O bonds and the four distinct bands at ~1723, 1735, 1763, and 1786 cm^−1^ indicate a variety of molecular locations in which the carbonyl unit has formed as a result of environmentally assisted biofilm processes. Using the intensity ratio of the 1725 to the 949 cm^−1^ bands the fraction of carbonyls, *S*, as a function of *S. sanguinis* fraction in coculture is given in (**f**), as computed according to the calibration procedures reported by Phillips et al. [51] and with statistical validation (*n* = 3; * → *p* < 0.05; ** → *p* < 0.01; *n.s.* = non-significant.

**Figure 10 ijms-24-06694-f010:**
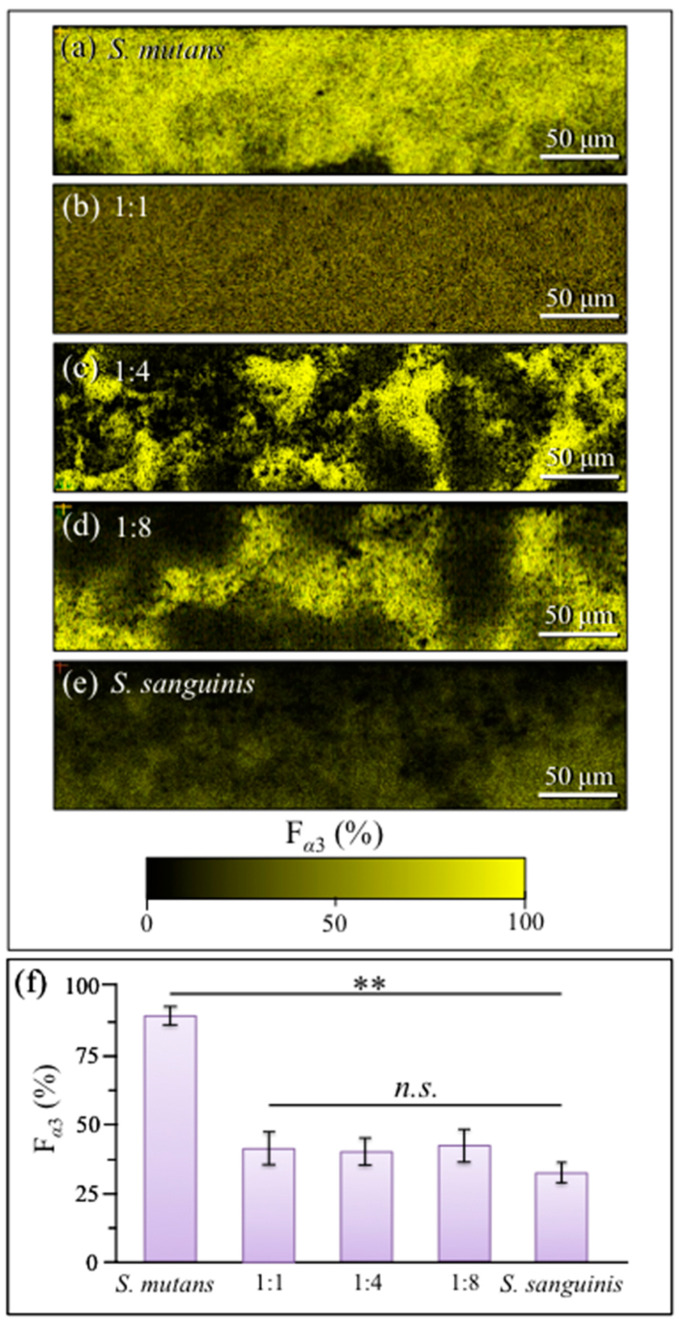
Raman maps of α − 1,3–glucan distribution in culture/cocultures of *S. mutans* and *S. sanguinis* with ratios of: (**a**) 1:0, (**b**) 1:1, (**c**) 1:4, (**d**) 1:8, and (**e**) 0:1; each map is comprehensive of ~10^6^ spectra collected with sub-micrometric spatial resolution. In (**f**), average values of α − 1,3–glucan fractions, *F*_α3_, are computed by averaging *R*^(1)^_α3/α6_ = *I*_520_/*I*_555_ ratios over entire Raman images (*n* = 3 for each culture/coculture), according to the calibration plot in Figure 3a. *F*_α3_ values showed agreement within ±10% with data in Figure 8f computed with the same algorithm (cf. also statistical validation; *n* = 3; ** → *p* < 0.01; *n.s.* = non significant). Raman maps were collected with focal plane on the sample surface.

**Figure 11 ijms-24-06694-f011:**
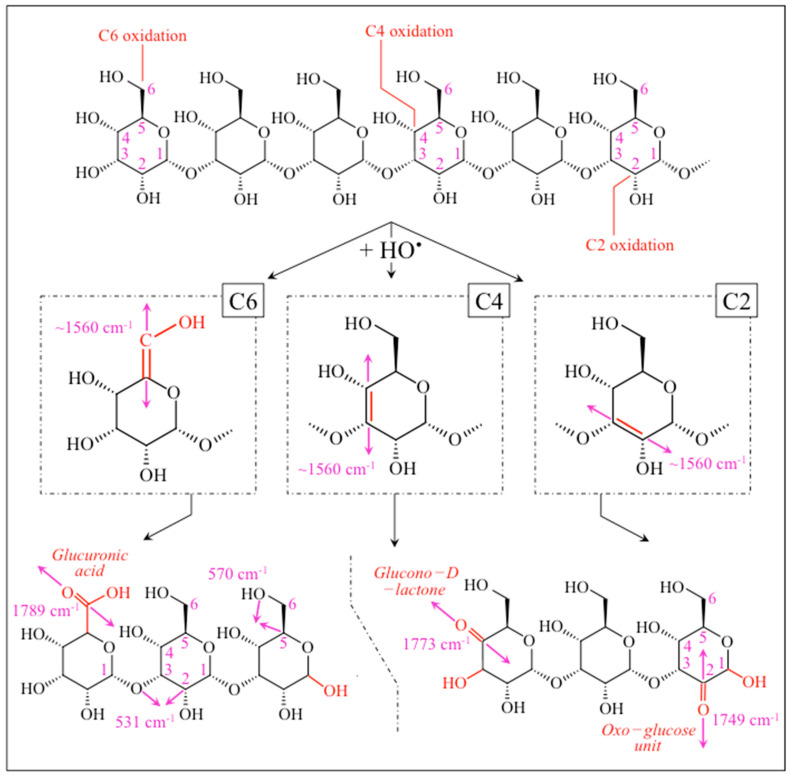
Draft of oxidized carbonyl sites (compiled according to Ref. [52]) as positioned in a glucan chain cleaved by HO• radicals; the respective (guessed) Raman frequencies are given in inset. The draft also envisages the possible formation of transient C=C bonds, which could contribute the intensity of the observed ~1560 cm^−1^ band.

**Figure 12 ijms-24-06694-f012:**
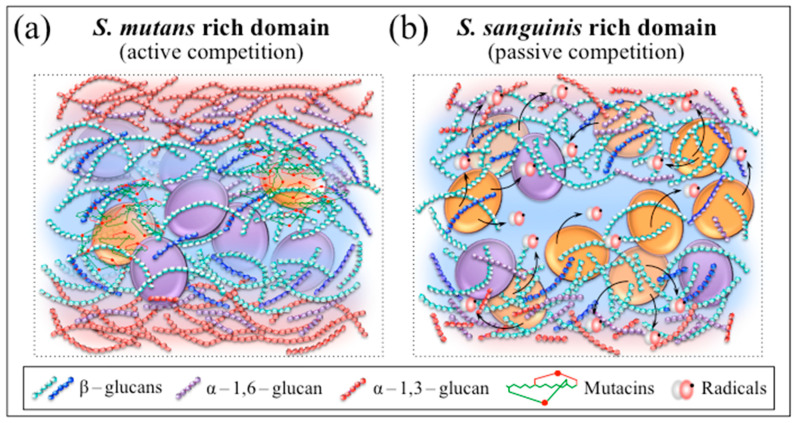
Schematic drafts of biofilm zones dominated by (**a**) *S. mutans* and (**b**) *S. sanguinis*. In the former case, the biofilm is rich in water-insoluble exopolysaccharides and *S. mutans* uses mutacins in the attempt to directly exterminate the *S. sanguinis*; in the latter case, *S. sanguinis* disrupts water-insoluble exopolysaccharides using oxygen radicals.

## Data Availability

Authors agree to make data and materials supporting the results or analyses presented in their paper available upon reasonable request.

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
