# Peer review of "In Situ Raman Analysis of Biofilm Exopolysaccharides Formed in Streptococcus mutans and Streptococcus sanguinis Commensal Cultures"

_ijms, 2023, doi:10.3390/ijms24076694_

Round 1

Reviewer 1 Report (New Reviewer)

The authors investigate an interesting competition mechanism between S. sanguinis and S. mutans to explore the in vitro biofilm interactions via Raman spectroscopy and imaging. By providing solid and abundant experimental data, they found that S. sanguinis followed a passive path for securing space competition by releasing oxygen radicals, while S. mutans adopted an active competitive strategy by secreting mutacins to directly inhibit of S. mutans growth. They show how Raman can be utilized to identify biofilm composition, and elucidate the mechanism of bacterial competition. I believe that this work deserves the publication in the international journal of molecular sciences. However, the following points have to be addressed before the publication.

1.       Could the authors properly arrange the pictures to avoid so much blank space?

2.       In addition, could the authors choose proper color or position to clearly show the letters or numbers in the figure 4 and figure 10?

3.       The authors must double check the typing issues in the context. For example, there is a typing problem in the line 606.

4.       The paragraph from line 392-404 actually is repeated.

Author Response

Reviewer 1:

The authors investigate an interesting competition mechanism between S. sanguinis and S. mutans to explore the in vitro biofilm interactions via Raman spectroscopy and imaging. By providing solid and abundant experimental data, they found that S. sanguinis followed a passive path for securing space competition by releasing oxygen radicals, while S. mutans adopted an active competitive strategy by secreting mutacins to directly inhibit of S. mutans growth. They show how Raman can be utilized to identify biofilm composition, and elucidate the mechanism of bacterial competition. I believe that this work deserves the publication in the international journal of molecular sciences. However, the following points have to be addressed before the publication.

  1. Could the authors properly arrange the pictures to avoid so much blank space?

We did our best to arrange the layout to avoid blank spaces. Nevertheless, we trust that the IJMS Editors will edit the final layout of the paper in the most efficient way, as done for our previous papers.

  1. In addition, could the authors choose proper color or position to clearly show the letters or numbers in the figure 4 and figure 10?

We edited the two figures indicated by the Reviewer in order to improve legend visibility.

  1. The authors must double check the typing issues in the context. For example, there is a typing problem in the line 606.

We had a double check on typing issues and fixed the English grammar (corrections in red ink).

  1. The paragraph from line 392-404 actually is repeated.

The repetition has been eliminated.

Reviewer 2 Report (New Reviewer)

In this manuscript (ijms-2257961), The authors used quantitative Raman spectroscopy and imaging to investigate the mechanism of competition/inhibition between two Streptococcus species. The performance of this method is interesting and satisfying, and it has great novelty. However, There are still some areas of the article that need to be optimized. Thus, I would like to recommend this manuscript to be published in International Journal of Molecular Sciences after major revisions.

Here are some reasons why we give this opinion:

1.     There is a great deal of repetition in the description of the vibrational modes corresponding to specific shifts in the Raman spectra, which makes the article too long. It is suggested that some of the vibrational modes corresponding to the positions of the Raman shifts can be summarized in a table, and some of the text can be deleted.

2.     The details of the tables in some articles need attention, e.g. Figure 9(c).

3.     It is recommended to simplify the article, the length of which is not commensurate with the workload.

Author Response

Reviewer 2:

In this manuscript (ijms-2257961), The authors used quantitative Raman spectroscopy and imaging to investigate the mechanism of competition/inhibition between two Streptococcus species. The performance of this method is interesting and satisfying, and it has great novelty. However, There are still some areas of the article that need to be optimized. Thus, I would like to recommend this manuscript to be published in International Journal of Molecular Sciences after major revisions.

Here are some reasons why we give this opinion:

  1. There is a great deal of repetition in the description of the vibrational modes corresponding to specific shifts in the Raman spectra, which makes the article too long. It is suggested that some of the vibrational modes corresponding to the positions of the Raman shifts can be summarized in a table, and some of the text can be deleted.

We have eliminated all repetitions (this point was also highlighted by the Reviewer 1). Thank you for noticing this problem. The main purpose of this paper is quantitative Raman spectrometry, therefore we consider appropriate to describe in detail the vibrational items of the spectra. We agree with the Reviewer that, for better clarity, it is better to add a Table with the wavenumbers and vibrational origins. We have done so by adding Table S-I in the Supplementary Information.

  1. The details of the tables in some articles need attention, e.g. Figure 9(c).

We have reviewed and corrected all figures.

  1. It is recommended to simplify the article, the length of which is not commensurate with the workload.

IJMS does not impose limitations on article length and this is one of the reasons we have chosen this journal for this study. We believe that there are many novel items in this article and, therefore, its length is appropriate. Eliminating parts related to spectroscopic explanations (as suggested by the Reviewer at the above point 1, will leave the Readers with insufficient information. We leave to the Editor the final decision.

Round 2

Reviewer 2 Report (New Reviewer)

The revised manuscript (ijms-2257961) has been improved. For the questions about this work, the authors have made careful revisions and supplements, and the responses are reasonable. Thus, I would like to recommend this paper to be published in International Journal of Molecular Sciences.

This manuscript is a resubmission of an earlier submission. The following is a list of the peer review reports and author responses from that submission.

Round 1

Reviewer 1 Report

In this study Raman spectroscopy was applied on biofilm of S. mutant and S. sanguinis in mono-and co-culture to probe the mechanism of their competition/coexistence in vitro. For results interpretation a library of reference Raman spectra was created and an algorithm was used for automatic sub-bands deconvolution based on the substances measured in the library. Based on this analysis the hypothesis was supported that S. sanguinis uses oxygen radicals for glucan cleavage against S. mutant in co-culture.

The results and their interpretation emerge major concern that require clarifications in order to be convincing.  It is not clear what exactly was measured by the investigators, only the extracellular matter or the entire biofilm including the bacterial cells. Raman band interpretation is confusing since the analysis focuses only on the presence of glucans and does ignore bands from other biomolecules also appearing at the same positions of the spectrum within overlapping bands. Since not all substances measured in the Raman library or all Raman bands of the spectra are mentioned in the manuscript, it is not clear if systematic mistakes were conducted that lead to selective interpretation of the results. The presence of some overlapping signals are mentioned in the discussion but these are only few from many other possible. In this context it is not clear how conclusions specific on glucan cleavage were drawn and all other biochemical processes were excluded. Kindly find my comments below:

Materials and Methods

·         „bacteria strains and culture conditions”: It is not clear to me why glucose was used in the standard curve for the determination of glucan concentrations. Kindly explain and/or add a reference to support this methodology. 

·         “Nuclear magnetic resonance analyses”: Kindly describe in the manuscript how the α-1,6- and α-1,3-glucans were isolated from the biofilm.

Results

·         In the abstract it is mentioned that this study focuses on biofilm exopolysaccharides. In the method section it is described that the planktonic bacteria were separated from the biofilm. In the manuscript you describe Raman measurements of isolated biofilm and focus specifically on glucans. However, it is not clear if the Raman measurements were performed only in the extracellular matrix or the entire biofilm, including the bacteria placed inside. In the discussion section, the interpretation of the results is focused on the extracellular matrix. Kindly clarify what exactly was measured to avoid confusion.

·         Raman markers of cocultures biofilm exopysacharides: If bacterial cells were present in the biofilm when applying Raman spectroscopy how was it guaranteed that the signals from these cells did not mix with the signals from the extracellular polysaccharides. The reason I’m asking this question is because the band at 892 cm-1, that has been assigned to glucan by the authors, can also be assigned to the (C - O - O) skeletal vibration of lipids (866-898 cm-1) in addition to the C–O–C glycosidic ring stretching vibration (~897 cm-1) of carbohydrates (doi.org/10.1002/jrs.4607, doi.org/10.1016/S0167-7012(02)00127-6). Also, the signals that were related to the sucrose of the culture medium (caption Figure 7) can also be assigned to other biomolecules. DNA backbone stretching vibration and C-C stretching vibration of proteins appear ~929cm-1 (doi.org/10.1016/j.optlastec.2018.06.034). Ring breathing vibration band of tyrosine rises at 854 cm-1 (doi.org/10.1002/jbio.201000020, doi.org/10.1002/jrs.2533) and the CH3 deformation band of protein and lipids  appears around 977 cm-1 (doi.org/10.1002/jbio.201000020). Especially the presence of tyrosine in the sample can be easily verified by a second band around 829 cm-1. Was such a band present the spectrum?

·         Figure 8: Also here, between 520-540cm-1 S–S stretching vibration and C-O-C glycosidic ring deformation vibration of proteins and carbohydrates appear (doi.org/10.1016/S0167-7012(02)00127-6, In Advances in Applied Microbiology, Academic Press: 2010; Vol. 70, pp. 153-186) how is this related to the biofilm?

·         Figure 9: Also here, the =O stretching vibration that appears in the spectral range 1727-1749cm-1 is characteristic for esters in lipids (doi.org/10.1002/jrs.4607, https://doi.org/10.1016/S0167-7012(02)00127-6) is this not an indicator that bacterial cells were measured?  

·         In lines 340, 436 and in the caption of Figure 9 the carbonyl bands are assigned ~1790 cm-1. Based on the low intensity of the bands after 1750cm-1 it looks more like noise rather than signal. Thus, these signals are not proper to draw conclusion on.

·         Stating that all the above Raman signals are only associated with glucans is not convincing since these bands can also include overlapping signals from other biomolecules. I would suggest to mention all the 40 compounds included into the Raman library and discuss this issue in the manuscript.

·         Figure 4/10: The scale in the images is too large to see identify the presence of only extracellular matter or also bacterial cells. Was the separation of the planktonic cells from the extracellular matter confirmed by culturing or any other method?

·         Line 458: at 1551 cm-1 the C-C stretching vibration of the pyrrole ring of tryptophan appears (doi.org/10.1002/bip.20489), why are you referring to a C=C vibration?

Author Response

Reviewer 1:

In this study Raman spectroscopy was applied on biofilm of S. mutant and S. sanguinis in mono-and co-culture to probe the mechanism of their competition/coexistence in vitro. For results interpretation a library of reference Raman spectra was created and an algorithm was used for automatic sub-bands deconvolution based on the substances measured in the library. Based on this analysis the hypothesis was supported that S. sanguinis uses oxygen radicals for glucan cleavage against S. mutant in co-culture.

The results and their interpretation emerge major concern that require clarifications in order to be convincing.  It is not clear what exactly was measured by the investigators, only the extracellular matter or the entire biofilm including the bacterial cells. Raman band interpretation is confusing since the analysis focuses only on the presence of glucans and does ignore bands from other biomolecules also appearing at the same positions of the spectrum within overlapping bands. Since not all substances measured in the Raman library or all Raman bands of the spectra are mentioned in the manuscript, it is not clear if systematic mistakes were conducted that lead to selective interpretation of the results. The presence of some overlapping signals are mentioned in the discussion but these are only few from many other possible. In this context it is not clear how conclusions specific on glucan cleavage were drawn and all other biochemical processes were excluded. Kindly find my comments below:

Materials and Methods

  • „bacteria strains and culture conditions”: It is not clear to me why glucose was used in the standard curve for the determination of glucan concentrations. Kindly explain and/or add a reference to support this methodology. 

The required information has been added in Section 4.1.

  • “Nuclear magnetic resonance analyses”: Kindly describe in the manuscript how the α-1,6- and α-1,3-glucans were isolated from the biofilm.

An extensive description of NMR analyses has been added in Section 4.2, as requested by the Reviewer. A new reference has also been added.

Results

  • In the abstract it is mentioned that this study focuses on biofilm exopolysaccharides. In the method section it is described that the planktonic bacteria were separated from the biofilm. In the manuscript you describe Raman measurements of isolated biofilm and focus specifically on glucans. However, it is not clear if the Raman measurements were performed only in the extracellular matrix or the entire biofilm, including the bacteria placed inside. In the discussion section, the interpretation of the results is focused on the extracellular matrix. Kindly clarify what exactly was measured to avoid confusion.

Thank you for pointing out this source of confusion in the manuscript. Separation was done only for preparing bacterial strains to be joined in cocultures with fixed population fractions, while Raman measurements were made after fixed time of coculture on the entire biofilm. This distinction has now been clarified in the abstract of the revised manuscript.

  • Raman markers of cocultures biofilm exopysacharides: If bacterial cells were present in the biofilm when applying Raman spectroscopy how was it guaranteed that the signals from these cells did not mix with the signals from the extracellular polysaccharides. The reason I’m asking this question is because the band at 892 cm-1, that has been assigned to glucan by the authors, can also be assigned to the (C - O - O) skeletal vibration of lipids (866-898 cm-1) in addition to the C–O–C glycosidic ring stretching vibration (~897 cm-1) of carbohydrates (doi.org/10.1002/jrs.4607, doi.org/10.1016/S0167-7012(02)00127-6). Also, the signals that were related to the sucrose of the culture medium (caption Figure 7) can also be assigned to other biomolecules. DNA backbone stretching vibration and C-C stretching vibration of proteins appear ~929cm-1(doi.org/10.1016/j.optlastec.2018.06.034). Ring breathing vibration band of tyrosine rises at 854 cm-1 (doi.org/10.1002/jbio.201000020, doi.org/10.1002/jrs.2533) and the CH3 deformation band of protein and lipids  appears around 977 cm-1 (doi.org/10.1002/jbio.201000020). Especially the presence of tyrosine in the sample can be easily verified by a second band around 829 cm-1. Was such a band present the spectrum?

We agree with the Reviewer on the fact that, at first glance, it is not possible to distinguish between overlapping contributions from different molecules. However, as described in Section 4.4, we used a machine-learning algorithm that picks elementary spectra from a library of Raman spectra and automatically determines the fractional contributions of different molecules to overlapping signals. We developed this machine-learning algorithm (described in previous literature [24,25]) to fit the experimental spectra by preserving relative intensities, spectral positions, and full-width at half-maximum values from each elementary compound (i.e., within ±3 cm-1; a boundary value selected by considering the resolution of the spectrometer and the possibility of slight alterations in molecular structure). Those mathematical constraints allowed univocal deconvolution of the experimental spectra. This procedure allowed us selecting the main contribution to the observed bands. Trying to give some example to explain how the algorithm works, if palmitic acid or TPA exists (doi.org/10.1002/jrs.4607), showing a band at ~898 cm-1, then the much stronger sharp band located at ~1132 cm-1 should also be clearly observed in Fig. 7. Concerning the carbohydrates, as shown in the review paper on Raman and infrared spectroscopy of carbohydrates (Wiercigroch, et al., Raman and infrared spectroscopy of carbohydrates: A review, Spectrochim. Acta Part A, 185(2017)317), most of the carbohydrates don’t show a band at around 897 cm-1, while in those carbohydrates that show its presence, this band is rather weak (e.g., L-(+)-arabinose). Similarly, for the case of tyrosine, as can be seen from Fig. 7, the band at ~829 cm-1 is quite weak for the co-cultured samples. On the contrary, as shown in our previous paper (DOI: 10.3389/fmicb.2021.769597), beta-glucan shows a relative strong Raman band at this position. In summary, machine learning allows to analyse the main contributions to the Raman bands, while neglecting the minor contributions.

         Figure 8: Also here, between 520-540cm-1 S–S stretching vibration and C-O-C glycosidic ring deformation vibration of proteins and carbohydrates appear (doi.org/10.1016/S0167-7012(02)00127-6, In Advances in Applied Microbiology, Academic Press: 2010; Vol. 70, pp. 153-186) how is this related to the biofilm?

The reply is based on the same logic explained above. In addition, although they show Raman bands in the range, in general the S-S disulfide stretching band also exhibits a sharp band located at ~510 cm-1 (Ozaki Y, Mizuno A, Itoh K, Iriyama K: Inter- and intramolecular disulfide bond formation and related structural changes in the lens proteins. A Raman spectroscopic study in vivo of lens aging, J Biol Chem 1987, 262:15545-15551.), which is negligible in Fig. 8. The band of C-O-C glycosidic ring deformation generally is located at around 540 cm-1, but in Fig. 8 the morphologies of the Raman spectra suggest the presence of peaks at 531 and 555 cm-1, and a minor/negligible contribution of this substance.

  • Figure 9: Also here, the =O stretching vibration that appears in the spectral range 1727-1749cm-1is characteristic for esters in lipids (doi.org/10.1002/jrs.4607, https://doi.org/10.1016/S0167-7012(02)00127-6) is this not an indicator that bacterial cells were measured?  

The Raman spectra of different kinds of lipid have been presented in a review paper (Czamara, et al., J. Raman Spectrosc. 2015, 46, 4). Yes, we agree that these signals could also be comprehensive of metabolites from cells. However, the observed progressive increase of such signals in cocultures increasingly rich in S. sanguinis (cf. Fig. 9) while significantly weaker in both S. mutans and S. sanguinis single cultures, supports the hypothesis that the main contribution to ester bands is a consequence of the interaction between the two bacterial species (from which the above interpretation).

  • In lines 340, 436 and in the caption of Figure 9 the carbonyl bands are assigned ~1790 cm-1. Based on the low intensity of the bands after 1750cm-1it looks more like noise rather than signal. Thus, these signals are not proper to draw conclusion on.

We have introduced in text at the end of page 11 that, as suggested by the Reviewer, some bands in the region 1750~1790 cm-1 have intensities comparable with noise and so it is difficult to assign to them a specific interpretation.

  • Stating that all the above Raman signals are only associated with glucans is not convincing since these bands can also include overlapping signals from other biomolecules. I would suggest to mention all the 40 compounds included into the Raman library and discuss this issue in the manuscript.

We believe that we have already answered to this question, which is basically the same as others above. As mentioned above, here we located the main contributions to the Raman bands through a machine learning algorithm, and neglected minor contributions to discuss compositional and structural alterations. This procedure has been extensively explained and applied in a number of our previous papers.

  • Figure 4/10: The scale in the images is too large to see identify the presence of only extracellular matter or also bacterial cells. Was the separation of the planktonic cells from the extracellular matter confirmed by culturing or any other method?

As explained at the beginning of this reply report, we have not separated bacteria and extracellular matrix in the Raman experiments. This is now better clarified in the revised manuscript.

  • Line 458: at 1551 cm-1the C-C stretching vibration of the pyrrole ring of tryptophan appears (doi.org/10.1002/bip.20489), why are you referring to a C=C vibration?

The pyrrole ring of tryptophan contains C=C-C=C bonds instead of single bonds of C-C. Indeed, the 1551 cm-1 band is attributed to ν(C=C) of tryptophan (Huang Z, McWilliams A, Lui H, McLean DI, Lam S, Zeng H: Near-infrared Raman spectroscopy for optical diagnosis of lung cancer, Int. J. Cancer 2003, 107: 1047-1052.).

Reviewer 2 Report

This study described a nondestructive Raman spectroscopic method to investigate the biofilm formed by Streptococcus mutans and Streptococcus sanguinis in-situ. The strategy of this research is smart and well-designed, which is of interest both from basic medicine and clinical medicine point of view, which is a great guide for researchers and clinicians. I recommend it for the publication before minor revise.

The main problem in this article is not concise enough

Author Response

Reviewer 2:

This study described a nondestructive Raman spectroscopic method to investigate the biofilm formed by Streptococcus mutans and Streptococcus sanguinis in-situ. The strategy of this research is smart and well-designed, which is of interest both from basic medicine and clinical medicine point of view, which is a great guide for researchers and clinicians. I recommend it for the publication before minor revise.

Thank you for your positive comments on our manuscript.

The main problem in this article is not concise enough

We tried to improve the clarity in the revised manuscript, also through the many comments made by all Reviewers. The content of the treated topic is very complex and studied by many researchers. We ask the Reviewer to understand that it is difficult to be concise on this topic without neglecting fundamental contributions and loosing important details.

Reviewer 3 Report

The paper describes the use of Raman spectroscopy for the analysis of biofilms formed in cocolonizations of Streptococcus sanguinis and Streptococcus mutans, responsible for the formation of caries in human oral cavity. The authors used their original methodology previously applied to the study of other biofilms. They concluded that Streptococcus sanguinis affected the impermeability of the biofilm constructed by Streptococcus mutans, possibly by oxidation of glucans. While the topic of the paper is of interest for this journal, the impact of the work comes down to the questions of how convincingly have the authors made the case that they are performing reliable Raman spectroscopy analysis and how the conclusions are supported by the experimental evidence. In this case, the answer to the second question is also tightly dependent on the answer to the first question. Since this study rely almost exclusively on Raman experiments, without any confirmation by other techniques, there are several issues that limit the power and conclusiveness of the data. Thus, I cannot recommend publication of this manuscript unless the authors address the following concerns:

1.          In Fig. 1 (b), it is hard to understand and judge the correct use of the baseline correction and the real signal-to-noise ratio of the experimental spectra. Improper baseline correction along with underestimation of the spectral noise can be deceiving, adding artifacts to the band analysis, especially in biological samples. The authors also used 532 nm laser source instead of red and near-infrared lasers, which are preferred to suppress fluorescence in biological samples. For instance, in Fig. 1 (b) at around 1200 cm-1, the minimum of the spectral line for α-1,6-glucan has been cut to zero upon baseline correction while for α-1,3-glucan has not. The band at 1200 cm-1 may be an artifact created by base line subtraction. The Authors should show in Supporting Information some examples of unfiltered experimental spectra prior to baseline subtraction, noise filtering and normalization of spectra.

2.          In Fig. 1 (b), the typical band of α-glucans at around 550 cm-1 in the reference spectrum is shifted to around 570 cm-1. The spectra for calibration in Fig. 2 (a) and the spectra of the biofilm in Fig. 6 and 8 show this band at the proper position (around 550 cm-1), although bands at 570 cm-1 are also observed. Is the reference spectrum correct? It is quite dubious that the authors reported a different reference spectrum for α-1,3-glucans in a previous paper (Ref. 25, doi: 10.3389/fmicb.2021.769597). The latter reference spectrum was also deconvoluted by a different set of Voigtian bands, which differ in total number of bands and some of their positions. This is the evidence that accurate quantitative analysis of molecules in biological samples by Raman spectroscopy is quite challenging and tricky.

3.          Page 4, line 131. The band of α-1,6-glucans at 520 cm-1 is assigned to C2-C3-OH vibration while in Ref. 32, the same authors assigned this band of α-1,6-glucans to in-plane ring vibration. Which is the correct assignment?

4.          The fractions in Fig. 4 are defined as S. mutans-to-S. sanguinis ratio. In lines 175-176 at page 6, 8:1 fraction should be the richest in S. mutans, not in S. sanguinis as stated by the authors. In the following explanation of the results of Fig. 4 (f), starting from line 186, it is unclear to me whether or not the authors are confusing which is the richest bacterium in the coculture, according to their definition of ratio. From my understanding, it appears that for whatever reason only small fractions of S. sanguinis enhanced biofilm formation in cocultures.

5.          Figure 5. Biofilms from S. sanguinis and the cocultures were not analyzed by NMR. NMR would confirm the results of Raman spectroscopy. The choice of analyzing only the biofilm of S. mutans is not scientifically sound.

6.          Line 209 at page 7. The Supporting Information file was not available to the reviewers.

7.          Line 222 at page 8. How was the thickness of the biofilm measured? 

8.          Figure 6 (a). The color map represents the distribution of Fα3. At z=0 and z=22 µm the color map looks similar (light green should be 60-70% according to the color bar). Instead, the calculated mean percentages in (b), (e), (d) and (g) are quite different: close to 100% for z = 0 and 50-60% for z = 22.  

9.          Spectra showed in Figs. 7-9 are the average of twenty spectra collected at different locations of the biofilms. What depth did the authors choose to acquire the spectra? This information is important following the results of Fig. 6.     

10.      Figures 7-10. For statistical analysis of more than two groups means, unpaired student’s t-test is not an appropriate choice. Authors should use ANOVA followed by a post-hoc test for comparison between two groups.

11.      In the plot of Fig. 7 (f), is Rα/β %? If so the intensity difference between the two bands is quite small and within the instrumental error of the Raman technique. Moreover, for S.sanguinis the mean Rα/β appears statistically different from 1:1, 4:1 and 8:1 (if error bars are st dev and n=3).

12.      Line 257 at page 9. The authors discuss the spectra of biofilms in the region 600-1200 cm-1. They assigned the band at 892 cm-1 to β-glucans. In the previous section the authors claimed that the glucans in the biofilm are α-1,3-glucan and α-1,6-glucan, according to the NMR analysis and the Raman results in the region 400-600 cm-1. These are two contradictory conclusions. Why NMR did not show β-glucans?

13.      If β-glucans are in the biofilm, reinterpretation of the Raman results from biofilms in the range 400-600 cm-1 (Figs. 6 and 8) is required. In fact, the reference spectrum of β-1,3-glucans reported by the authors in a previous study (Ref. 25, doi: 10.3389/fmicb.2021.769597) clearly shows some Raman bands overlapping the band of α-1,3-glucan at 520 cm-1, which was used by the authors for quantitative analysis of Rα in this paper.

14.      Figure 8. The authors affirm in the caption that the bands at 531 and 572 cm-1 are only expected in α-1,3-glucans. Nonetheless, in the reference spectrum of β-1,3-glucans of Ref. 25, the same authors used several bands to deconvolute the spectral line in that region. Among those bands there are also two peaks at around 530 and 570 cm-1.

15.      All in all, comments 1, 2, 12, 13, 14 are the main concerns that demand for careful reconsideration of the spectral interpretation and band fitting, especially in the range 400-600 cm-1. Although the strive for quantification proposed in this study is undeniable, the reliability of the machine learning algorithm for spectral deconvolution conceived by the authors is yet to be proved.

16.      Fig. 10. As in comment #9, information about the depth of the laser focal plane in the biofilm during Raman acquisition should be reported in the caption or in the text.

17.      Fig. 10 (b). What is the yellow square under the scale bar? Because of its regular shape, it does not seem a feature attributable to bacteria.

18.      Line 513 at page 16. As in comment #4, I am still confused by the definition of S. mutans-to-S. sanguinis ratio given by the authors. Figure 10(d) shows images of 8:1 ratio. Is S. mutans under conditions of striking minority?

19.      Line 544 at page 17. As in the previous comment, does fraction of S. sanguinis up to 87.5% refers to the coculture 8:1? As far as I understood, it should be 87.5% S. mutans. 

20.      Lines 714-715 at page 20. Reference spectra for α-glucans were acquired using 514 nm laser source with nominal power of 200 mW. This is too powerful for analyses of biomolecules. 20-fold higher than the laser power used by the authors for the analysis of biofilms. Reference spectra should be acquired using the same experimental conditions used in biofilms, avoiding laser-induced damage.

Author Response

Reviewer 3:

The paper describes the use of Raman spectroscopy for the analysis of biofilms formed in cocolonizations of Streptococcus sanguinis and Streptococcus mutans, responsible for the formation of caries in human oral cavity. The authors used their original methodology previously applied to the study of other biofilms. They concluded that Streptococcus sanguinis affected the impermeability of the biofilm constructed by Streptococcus mutans, possibly by oxidation of glucans. While the topic of the paper is of interest for this journal, the impact of the work comes down to the questions of how convincingly have the authors made the case that they are performing reliable Raman spectroscopy analysis and how the conclusions are supported by the experimental evidence. In this case, the answer to the second question is also tightly dependent on the answer to the first question. Since this study rely almost exclusively on Raman experiments, without any confirmation by other techniques, there are several issues that limit the power and conclusiveness of the data. Thus, I cannot recommend publication of this manuscript unless the authors address the following concerns:

  1. In Fig. 1 (b), it is hard to understand and judge the correct use of the baseline correction and the real signal-to-noise ratio of the experimental spectra. Improper baseline correction along with underestimation of the spectral noise can be deceiving, adding artifacts to the band analysis, especially in biological samples. The authors also used 532 nm laser source instead of red and near-infrared lasers, which are preferred to suppress fluorescence in biological samples. For instance, in Fig. 1 (b) at around 1200 cm-1, the minimum of the spectral line for α-1,6-glucan has been cut to zero upon baseline correction while for α-1,3-glucan has not. The band at 1200 cm-1may be an artifact created by base line subtraction. The Authors should show in Supporting Information some examples of unfiltered experimental spectra prior to baseline subtraction, noise filtering and normalization of spectra.

The mathematical procedure leading to the final spectrum is not something that can be clearly observed by naked eyes. We applied standard procedures for background subtraction, which take into account the entire spectrum, not a single location, and this procedure can sometimes cause little distortions around the edges of a certain spectroscopic window. More details about the way baseline was subtracted are given in Section. 4.4, but the same method was used for all samples, in order to further improve reliability, in particular for comparative results. Please note that despite reducing fluorescence, NIR lasers give really weak Raman signals, in particular for thin biological samples. The 532 nm laser was the best in giving us the highest signal-to-noise ratio (between NIR, 488 nm, 514 nm and 532 nm, the four sources we commonly use).

  1. In Fig. 1 (b), the typical band of α-glucans at around 550 cm-1in the reference spectrum is shifted to around 570 cm-1. The spectra for calibration in Fig. 2 (a) and the spectra of the biofilm in Fig. 6 and 8 show this band at the proper position (around 550 cm-1), although bands at 570 cm-1 are also observed. Is the reference spectrum correct? It is quite dubious that the authors reported a different reference spectrum for α-1,3-glucans in a previous paper (Ref. 25, doi: 10.3389/fmicb.2021.769597). The latter reference spectrum was also deconvoluted by a different set of Voigtian bands, which differ in total number of bands and some of their positions. This is the evidence that accurate quantitative analysis of molecules in biological samples by Raman spectroscopy is quite challenging and tricky.

 The location of bands and their morphology greatly depend on the chemical state and link with the neighboring molecules (molecular pairing). This might be different even in the same sample at different locations, in particular for complex biological samples. For this reason, we have built a machine-learning algorithm that gives us the most probable deconvolution based on strict mathematical constraints. We hypothesized that the band is related to the same vibration (and the same molecule) based on proximity, after checking for other possible sources. It might be a weak point in the data interpretation, but in absence of better references this speculation sounds solid enough to us.

  1. Page 4, line 131. The band of α-1,6-glucans at 520 cm-1is assigned to C2-C3-OH vibration while in Ref. 32, the same authors assigned this band of α-1,6-glucans to in-plane ring vibration. Which is the correct assignment?

Since this band is almost negligible in α-1,3-glucan, we considered it should arise from a vibration that is not present in the material, i.e., the C2-C3-OH vibration is more reasonable. As the reviewer knows, vibrations at the same spectral position can have multiple interpretations.

  1. The fractions in Fig. 4 are defined as S. mutans-to-S. sanguinisratio. In lines 175-176 at page 6, 8:1 fraction should be the richest in S. mutans, not in S. sanguinis as stated by the authors. In the following explanation of the results of Fig. 4 (f), starting from line 186, it is unclear to me whether or not the authors are confusing which is the richest bacterium in the coculture, according to their definition of ratio. From my understanding, it appears that for whatever reason only small fractions of S. sanguinis enhanced biofilm formation in cocultures.

 As stated in Materials and Method, it should be “S. sanguinis-to-S. mutans concentration ratios”.

  1. Figure 5. Biofilms from S. sanguinisand the cocultures were not analyzed by NMR. NMR would confirm the results of Raman spectroscopy. The choice of analyzing only the biofilm of S. mutans is not scientifically sound.

  1. sanguinis alone does not produce biofilm (cf. Fig. 4(e)), so we could not perform NMR.

  1. Line 209 at page 7. The Supporting Information file was not available to the reviewers.

We don’t know why the Supporting Information was not made available to the reviewer. We can only upload it again.

  1. Line 222 at page 8. How was the thickness of the biofilm measured? 

It was measured by laser microscopy.

  1. Figure 6 (a). The color map represents the distribution of Fα3. At z=0 and z=22 µm the color map looks similar (light green should be 60-70% according to the color bar). Instead, the calculated mean percentages in (b), (e), (d) and (g) are quite different: close to 100% for z = 0 and 50-60% for z = 22.  

 We understand there can be differences in the gradation of green due to the conversion of the figure. The scale and the map were obtained using different software and this might have influenced the visual result. We apologize about it.

  1. Spectra showed in Figs. 7-9 are the average of twenty spectra collected at different locations of the biofilms. What depth did the authors choose to acquire the spectra? This information is important following the results of Fig. 6.     

Focus is placed on the sample surface at each location. Added to the caption.

  1. Figures 7-10. For statistical analysis of more than two groups means, unpaired student’s t-test is not an appropriate choice. Authors should use ANOVA followed by a post-hoc test for comparison between two groups.

  1. In the plot of Fig. 7 (f), is Rα/β%? If so the intensity difference between the two bands is quite small and within the instrumental error of the Raman technique. Moreover, for S.sanguinis the mean Rα/β appears statistically different from 1:1, 4:1 and 8:1 (if error bars are st dev and n=3).

These are average spectra. We indeed used ANOVA, the previous claim about using unpaired student’s t-test was a mistake on our side.

  1. Line 257 at page 9. The authors discuss the spectra of biofilms in the region 600-1200 cm-1. They assigned the band at 892 cm-1to β-glucans. In the previous section the authors claimed that the glucans in the biofilm are α-1,3-glucan and α-1,6-glucan, according to the NMR analysis and the Raman results in the region 400-600 cm-1. These are two contradictory conclusions. Why NMR did not show β-glucans?

We do not have data to discuss about differences in sensitivity to detect the low content of β-glucan. Regarding data from different approaches, we can only show them as they are collected.

  1. If β-glucans are in the biofilm, reinterpretation of the Raman results from biofilms in the range 400-600 cm-1(Figs. 6 and 8) is required. In fact, the reference spectrum of β-1,3-glucans reported by the authors in a previous study (Ref. 25, doi: 10.3389/fmicb.2021.769597) clearly shows some Raman bands overlapping the band of α-1,3-glucan at 520 cm-1, which was used by the authors for quantitative analysis of Rα in this paper.

Data are comprehensive of both biofilm and cells’ structure.

  1. Figure 8. The authors affirm in the caption that the bands at 531 and 572 cm-1are only expected in α-1,3-glucans. Nonetheless, in the reference spectrum of β-1,3-glucans of Ref. 25, the same authors used several bands to deconvolute the spectral line in that region. Among those bands there are also two peaks at around 530 and 570 cm-1.

The statement of “the bands at 531 and 572 cm-1 are only expected in α-1,3-glucans” are correct for comparing only α-1,3-glucans and α-1,6-glucans.

  1. All in all, comments 1, 2, 12, 13, 14 are the main concerns that demand for careful reconsideration of the spectral interpretation and band fitting, especially in the range 400-600 cm-1. Although the strive for quantification proposed in this study is undeniable, the reliability of the machine learning algorithm for spectral deconvolution conceived by the authors is yet to be proved.

We proposed a method for quantification based on our algorithm and preliminary results, but we agree with the reviewer about the possibility to further improve and strengthen the approach. We propose to the reviewer to see this paper as a first step in the direction of Raman-based quantification, and not a final, comprehensive and consolidated procedure. In this paper, we presented our results and speculations, which may look weak due to the lack of reliable references. As the interest in this field of research increases, our primitive methods will surely reach a higher level of maturity, in or future work or in that of other researchers.

  1. Fig. 10. As in comment #9, information about the depth of the laser focal plane in the biofilm during Raman acquisition should be reported in the caption or in the text.

 Focus is placed on the sample surface at each location. Added to the caption.

  1. Fig. 10 (b). What is the yellow square under the scale bar? Because of its regular shape, it does not seem a feature attributable to bacteria.

 The problem has been fixed.

  1. Line 513 at page 16. As in comment #4, I am still confused by the definition of S. mutans-to-S. sanguinis ratio given by the authors. Figure 10(d) shows images of 8:1 ratio. Is S. mutansunder conditions of striking minority?

The problem has been fixed. S. mutans is under conditions of striking minority. 

  1. Line 544 at page 17. As in the previous comment, does fraction of S. sanguinisup to 87.5% refers to the coculture 8:1? As far as I understood, it should be 87.5% S. mutans

 Corrected, as in the previous comment.

  1. Lines 714-715 at page 20. Reference spectra for α-glucans were acquired using 514 nm laser source with nominal power of 200 mW. This is too powerful for analyses of biomolecules. 20-fold higher than the laser power used by the authors for the analysis of biofilms. Reference spectra should be acquired using the same experimental conditions used in biofilms, avoiding laser-induced damage.

Misprint corrected.

Reviewer 4 Report

The paper "In situ Raman analysis of biofilm ..." by Pezzotti et al. deals out an investigation on the mechanisms of competition/coexistence between Streptococcus sanguinis and Streptococcus mutnas by means of quantitative raman spectrscopy and imaging in vivo by quantifying glucan metabolites in live biofilms.

The paper is very interesting, well organized, the findings are highly convincing and well written. The methodology of applying Raman spectroscopy and imaging is rigorous.

It adds new knowledge to several topics,anging from biology to biochemistry, biomaterials etc., where Raman spectroscopy can play a leading role to quantify metabolites in vivo in biofilms.

Before expressing its acceptance which is in any case the final desire of my review, I would like to point out a few points to be clarified:

1) On page 8 (on total pages 27) about figure 6, the authors declare that the map ia acquired with spatial resolution of 670nm, and focal plane displacements of nearly 100nm. I suggest to authors to give a justification of such values.

2) On page 20, section on Raman library: the first sentence appears to be without a verbal predicate.

Anyway, the paper is really excellent and I suggest a quick publication.

Author Response

Reviewer 4:

The paper "In situ Raman analysis of biofilm ..." by Pezzotti et al. deals out an investigation on the mechanisms of competition/coexistence between Streptococcus sanguinis and Streptococcus mutnas by means of quantitative raman spectrscopy and imaging in vivo by quantifying glucan metabolites in live biofilms.

The paper is very interesting, well organized, the findings are highly convincing and well written. The methodology of applying Raman spectroscopy and imaging is rigorous.

It adds new knowledge to several topics,anging from biology to biochemistry, biomaterials etc., where Raman spectroscopy can play a leading role to quantify metabolites in vivo in biofilms.

Before expressing its acceptance which is in any case the final desire of my review, I would like to point out a few points to be clarified:

1) On page 8 (on total pages 27) about figure 6, the authors declare that the map ia acquired with spatial resolution of 670nm, and focal plane displacements of nearly 100nm. I suggest to authors to give a justification of such values.

We added the information related to the in depth resolution, while regarding the lateral one, we got the number from the Raman manufacturer.

2) On page 20, section on Raman library: the first sentence appears to be without a verbal predicate.

Corrected. Thank you.

Anyway, the paper is really excellent and I suggest a quick publication.

Thank you for your positive comments.

Round 2

Reviewer 1 Report

Required changes have been performed. Manuscript is ready to be published in current form. 

Reviewer 3 Report

The Authors marginally answered to my comments. The manuscript is basically the same version I reviewed a couple of weeks ago. The Authors acknowledged the speculative nature of their method, claiming that their speculations are solid enough to them. Science should be based on solid and supported hypotheses rather than a solid guess (speculation). Unfortunately, even accepting this paper as speculative science, the Authors’ answers to some of my comments proved that the Raman data analysis is biased and unreliable.  I cannot recommend publication on the International Journal of Molecular Science. The main concerns can be found in my following answers to some of the Authors’ reply:

1.          R: In Fig. 1 (b), it is hard to understand and judge the correct use of the baseline correction and the real signal-to-noise ratio of the experimental spectra. Improper baseline correction along with underestimation of the spectral noise can be deceiving, adding artifacts to the band analysis, especially in biological samples. The authors also used 532 nm laser source instead of red and near-infrared lasers, which are preferred to suppress fluorescence in biological samples. For instance, in Fig. 1 (b) at around 1200 cm-1, the minimum of the spectral line for α-1,6-glucan has been cut to zero upon baseline correction while for α-1,3-glucan has not. The band at 1200 cm-1may be an artifact created by base line subtraction. The Authors should show in Supporting Information some examples of unfiltered experimental spectra prior to baseline subtraction, noise filtering and normalization of spectra.

A: The mathematical procedure leading to the final spectrum is not something that can be clearly observed by naked eyes. We applied standard procedures for background subtraction, which take into account the entire spectrum, not a single location, and this procedure can sometimes cause little distortions around the edges of a certain spectroscopic window. More details about the way baseline was subtracted are given in Section. 4.4, but the same method was used for all samples, in order to further improve reliability, in particular for comparative results. Please note that despite reducing fluorescence, NIR lasers give really weak Raman signals, in particular for thin biological samples. The 532 nm laser was the best in giving us the highest signal-to-noise ratio (between NIR, 488 nm, 514 nm and 532 nm, the four sources we commonly use).

         R: The Authors did not show in Supporting Information examples of unprocessed data as suggested by the Reviewer. This is a simple request that does not require much effort to be fulfilled. The reviewer and certainly many other colleagues are perfectly capable to judge the influence of noise and baseline correction even by the naked eye. My request is further supported by the comments of Reviewer #1, who correctly interpreted the spectra showed in fig. 9 as affected by noise in the region 1750-1800 cm-1, even if the spectra are presented by the Authors in the processed and filtered form. The Authors are free to deny raw data sharing after publication, even if it is a suspicious practice. Nonetheless, the reviewer’s duty is to verify the authenticity and the consistency of the data presented before publication.

2.          R: In Fig. 1 (b), the typical band of α-glucans at around 550 cm-1 in the reference spectrum is shifted to around 570 cm-1. The spectra for calibration in Fig. 2 (a) and the spectra of the biofilm in Fig. 6 and 8 show this band at the proper position (around 550 cm-1), although bands at 570 cm-1 are also observed. Is the reference spectrum correct? It is quite dubious that the authors reported a different reference spectrum for α-1,3-glucans in a previous paper (Ref. 25, doi: 10.3389/fmicb.2021.769597). The latter reference spectrum was also deconvoluted by a different set of Voigtian bands, which differ in total number of bands and some of their positions. This is the evidence that accurate quantitative analysis of molecules in biological samples by Raman spectroscopy is quite challenging and tricky.

A: The location of bands and their morphology greatly depend on the chemical state and link with the neighboring molecules (molecular pairing). This might be different even in the same sample at different locations, in particular for complex biological samples. For this reason, we have built a machine-learning algorithm that gives us the most probable deconvolution based on strict mathematical constraints. We hypothesized that the band is related to the same vibration (and the same molecule) based on proximity, after checking for other possible sources. It might be a weak point in the data interpretation, but in absence of better references this speculation sounds solid enough to us.

R: In this particular case, shift of about 20 cm-1 in glucans is surprising. It is usually observed in massive structural rearrangement such as change of secondary structure in proteins (see difference in Amide bands of beta sheets and alpha helices). I thought the reference spectra in Fig. 1 (b) were somehow wrongly reported. Instead, in the answer to my comment, the Authors acknowledged my claim that accurate quantitative analysis of molecules in biological samples by Raman spectroscopy is quite challenging and tricky.

The machine-learning algorithm used by the authors cannot overcome this issue. In fact, the answer to my comment clearly confirms (unwittingly??) the reason why the procedure is intrinsically unreliable. In fact, the uncertainty is in the reference spectra of the library, as candidly acknowledged by the Authors. As also pointed out by Reviewer #1, cells embedded into the biofilm make things worse.

The source code of the program conceived by the Authors to implement the machine-learning algorithm has never been disclosed by the Authors.  Only its basic principle has been described here and also in other previous papers by the Authors. Nonetheless, even from the small amount of information available here and considering the Reviewers’ answer, I have plenty of reasons to affirm that the results of the fitting are biased. In fact, in section 4.4, the Authors explain (and I quote): “the algorithm located the best-fitting combination of the experimental spectra by preserving relative intensities, spectral positions, and full-width at half-maximum values from each elementary compound (i.e., within ±3 cm-1; a boundary value selected by considering the resolution of the spectrometer and the possibility of slight alterations in molecular structure).” In this paper the typical band of α-glucans of the reference spectrum in Fig.  1b (i.e., the library) is located at around 570 cm-1. How could the machine-learning algorithm locate this band at 550 cm-1 in the experimental spectra of mixtures of elementary compounds and the biofilms (Fig. 2, 6 and 8) if the band position can vary within ±3 cm-1? The strict mathematical constraints claimed by the Authors are not met by the program.

I suggest the Authors to share the source code of their program. It is common practice to validate new computational programs in specialized journals (e.g., Plos computational biology).

4.          R: The fractions in Fig. 4 are defined as S. mutans-to-S. sanguinis ratio. In lines 175-176 at page 6, 8:1 fraction should be the richest in S. mutans, not in S. sanguinis as stated by the authors. In the following explanation of the results of Fig. 4 (f), starting from line 186, it is unclear to me whether or not the authors are confusing which is the richest bacterium in the coculture, according to their definition of ratio. From my understanding, it appears that for whatever reason only small fractions of S. sanguinis enhanced biofilm formation in cocultures.

A: As stated in Materials and Method, it should be “S. sanguinis-to-S. mutans concentration ratios”.

          R: In Materials and method is indeed reported “S. sanguinis-to-S. mutans concentration ratios” but in Results and discussion is still S. mutans-to-S.sanguinis. The ratio in the figures has been changed according to S. mutans-to-S.sanguinis, which is fine. Please keep the name of the ratio in Materials and method consistent with the one used in Results, discussion and figures.

5.          R: Figure 5. Biofilms from S. sanguinis and the cocultures were not analyzed by NMR. NMR would confirm the results of Raman spectroscopy. The choice of analyzing only the biofilm of S. mutans is not scientifically sound.

A: sanguinis alone does not produce biofilm (cf. Fig. 4(e)), so we could not perform NMR.

          R: S. sanguinis produce biofilm but in Fig. 4 (e) the amount is negligible compared to s. mutants after 24 h. Author should try at least one-week incubation and preparing more dishes to collect sufficient amount of biofilm for NMR analysis. The requested task is not impossible. Besides, what about the other cases of cocultures that I also requested? These cases show abundant biofilm. The Authors ignored my comment.

6.          R: Line 222 at page 8. How was the thickness of the biofilm measured?

          A: It was measured by laser microscopy.

          R: Why this information is not added to the revised manuscript? The Authors still ignored the Reviewer’s comment.

7.          R: Figure 6 (a). The color map represents the distribution of Fα3. At z=0 and z=22 µm the color map looks similar (light green should be 60-70% according to the color bar). Instead, the calculated mean percentages in (b), (e), (d) and (g) are quite different: close to 100% for z = 0 and 50-60% for z = 22.

A: We understand there can be differences in the gradation of green due to the conversion of the figure. The scale and the map were obtained using different software and this might have influenced the visual result. We apologize about it.

          R: Figure and scale should be prepared using the same software. I still do not see difference between z=0 and z= 22 in fig. 6 (a). Are the Authors interested in preparing an improved manuscript? There is no evidence of 93% or 97% of Fα3 at z=0 in fig. 6(a). Actually I see darker green spots (tiny) close to the line of z=22, that should be 50-60%. The dark green of 90-100% is not observed in the map. Poor image preparation.

10.      R: Figures 7-10. For statistical analysis of more than two groups means, unpaired student’s t-test is not an appropriate choice. Authors should use ANOVA followed by a post-hoc test for comparison between two groups.

11.      R:  In the plot of Fig. 7 (f), is Rα/β%? If so the intensity difference between the two bands is quite small and within the instrumental error of the Raman technique. Moreover, for S.sanguinis the mean Rα/β appears statistically different from 1:1, 4:1 and 8:1 (if error bars are st dev and n=3).

A: These are average spectra. We indeed used ANOVA, the previous claim about using unpaired student’s t-test was a mistake on our side.

          R: The new reference 106 added by the Authors, which is about the use of t-test for partially paired data in clinical studies is inappropriate and unnecessary. This paper is about clinical studies that compare 2 groups of patients before and after treatment (t-test and its variants are appropriate in this case). Although the Authors report the use of ANOVA in the revised manuscript, the information of statistical analysis is still incomplete and confusing. ANOVA reveals only the existence of significant difference between some of the tested groups (more than 2). ANOVA does not uncover specific difference between groups. As I suggested in the comment, the authors should specify which post-hoc test was used to compare each pair of groups. I wonder how the authors calculated the p values to compare 2 groups, which led them to add the asterisks to the plots of figures 4-7-8-9-10.

          The authors did not answer to the comment 11, about Rα/β in fig. 7(f). By logic, I think the ratio should not be %. Please fix it in the axis name of fig. 7 (f).

12.      R: Line 257 at page 9. The authors discuss the spectra of biofilms in the region 600-1200 cm-1. They assigned the band at 892 cm-1 to β-glucans. In the previous section the authors claimed that the glucans in the biofilm are α-1,3-glucan and α-1,6-glucan, according to the NMR analysis and the Raman results in the region 400-600 cm-1. These are two contradictory conclusions. Why NMR did not show β-glucans?

A: We do not have data to discuss about differences in sensitivity to detect the low content of β-glucan. Regarding data from different approaches, we can only show them as they are collected.

          R: The content of β-glucans is not low according to Raman spectroscopy. The Authors present the ratio Rα/β in Fig. 7. The intensities of the two bands from α and β glucans are pretty much similar and in the same order of magnitude.

          If not the authors of the manuscript, who should discuss the results reported in this paper? There is no difference in sensitivity for α and β glucans using NMR; they can be both detected in glucan mixtures (see for example 10.3978/j.issn.2305-5839.2014.02.07). Besides, the authors of a manuscript cannot merely report some of the data as collected without taking responsibility for their interpretation, comparison and accuracy. The reviewer is certainly happy if his doubts are cleared up with convincing arguments.

13.      R: If β-glucans are in the biofilm, reinterpretation of the Raman results from biofilms in the range 400-600 cm-1(Figs. 6 and 8) is required. In fact, the reference spectrum of β-1,3-glucans reported by the authors in a previous study (Ref. 25, doi: 10.3389/fmicb.2021.769597) clearly shows some Raman bands overlapping the band of α-1,3-glucan at 520 cm-1, which was used by the authors for quantitative analysis of Rα in this paper.

A: Data are comprehensive of both biofilm and cells’ structure.

          R: The cells embedded into the biofilm are another source of uncertainty in the characterization of the Raman spectra (as also pointed out by Reviewer #1). The authors not only neglected the influence of β glucans in the range 400-600 but they acknowledged in this answer that they neglected possible contributions of intracellular biomolecules and structures. The Authors replied to Reviewer #1 saying that in general the S-S disulfide stretching band also exhibits a sharp band located at ~510 cm-1. I believe this band is the same one referred by Reviewer #1 (i.e., S-S stretching). This band is not necessarily sharp and located at 510 cm-1. It can be found at 520-540 cm-1 as reported by Reviewer #1 and broaden about 10-20 cm-1.

14.      R: Figure 8. The authors affirm in the caption that the bands at 531 and 572 cm-1are only expected in α-1,3-glucans. Nonetheless, in the reference spectrum of β-1,3-glucans of Ref. 25, the same authors used several bands to deconvolute the spectral line in that region. Among those bands there are also two peaks at around 530 and 570 cm-1.

A: The statement of “the bands at 531 and 572 cm-1 are only expected in α-1,3-glucans” are correct for comparing only α-1,3-glucans and α-1,6-glucans.

          R: I disagree. What about the band assigned to β-1,3-glucans in Ref. 25? The Raman markers for α – and β – glucans at 948 and 892 cm-1 are comparable in Fig. 7. It means that the bands of β – glucans in the region 500-570 cm-1 from Ref. 25 should have been located in the experimental spectra of biofilms by the machine learning algorithm. The Authors did not respond to my query, but they only neglected in this manuscript the existence of bands previously reported in their studies.

15.      R: All in all, comments 1, 2, 12, 13, 14 are the main concerns that demand for careful reconsideration of the spectral interpretation and band fitting, especially in the range 400-600 cm-1. Although the strive for quantification proposed in this study is undeniable, the reliability of the machine learning algorithm for spectral deconvolution conceived by the authors is yet to be proved.

A: We proposed a method for quantification based on our algorithm and preliminary results, but we agree with the reviewer about the possibility to further improve and strengthen the approach. We propose to the reviewer to see this paper as a first step in the direction of Raman-based quantification, and not a final, comprehensive and consolidated procedure. In this paper, we presented our results and speculations, which may look weak due to the lack of reliable references. As the interest in this field of research increases, our primitive methods will surely reach a higher level of maturity, in or future work or in that of other researchers.

R: The Authors agree with me but the opinion reported in this answer is not mentioned in the manuscript, but is exclusively shared with the Reviewer. Overall, I found only two modifications of the manuscript based on my comments (all minor modification, such as change 200 mW to 20 mW).

The method proposed by the Authors requires validation rather than further improvement. I strongly believe that the machine learning algorithm did not work correctly.

First, it should be impossible for the algorithm to move the α-glucans reference bands from 570 cm-1 to the position at 550 cm-1, as instead shown in the experimental spectra of the biofilms. Second, the algorithm is not supposed to use the reference spectrum of β- glucans to fit the band at 892 cm-1 and neglect the other bands of β- glucans in the range 500-570- cm-1.
